# Dissecting Query-Key Interaction in Vision Transformers

**Xu Pan**[1,2]   **Aaron Philip** [3]   **Ziqian Xie** [4]   **Odelia Schwartz** [1]
[1]University of Miami   [2]Harvard University   [3]Michigan State University
[4]University of Texas Health Science Center at Houston
xupan@fas.harvard.edu   philipaa@msu.edu
ziqian.xie@uth.tmc.edu   odelia@cs.miami.edu

## Abstract

Self-attention in vision transformers is often thought to perform perceptual grouping where tokens attend to other tokens with similar embeddings, which could correspond to semantically similar features of an object. However, attending to dissimilar tokens can be beneficial by providing contextual information. We propose to analyze the query-key interaction by the singular value decomposition of the interaction matrix (i.e. $\mathbf{W}_q^\top \mathbf{W}_k$). We find that in many ViTs, especially those with classification training objectives, early layers attend more to similar tokens, while late layers show increased attention to dissimilar tokens, providing evidence corresponding to perceptual grouping and contextualization, respectively. Many of these interactions between features represented by singular vectors are interpretable and semantic, such as attention between relevant objects, between parts of an object, or between the foreground and background. This offers a novel perspective on interpreting the attention mechanism, which contributes to understanding how transformer models utilize context and salient features when processing images.

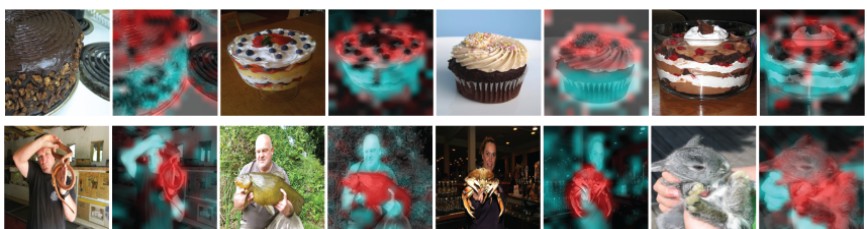

Figure 1: We propose a new way to study query-key interactions via the singular value decomposition of the query-key interaction matrix. Many of the modes (i.e. pairs of singular vectors corresponding to the query and the key respectively), are semantically interpretable. Two example modes are shown. Top row: ViT layer 8 head 7 mode 2. Bottom row: DINO layer 8 head 9 mode 2. The red channel indicates the projection value of embedding onto the left singular vector which corresponds to the query; the cyan channel indicates the projection value of embedding onto the right singular vector which corresponds to the key.

## 1   Introduction

Vision transformers (ViTs) are a family of models that have significantly advanced the computer vision field in recent years [14]. The core computation of ViTs, self-attention, is designed to promote interactions between tokens corresponding to relevant image features [14]. But this mechanism

38th Conference on Neural Information Processing Systems (NeurIPS 2024).

has different interpretations with open questions such as what "relevant" refers to. Some interpret "relevant" as tokens within the same object. Highlighting objects in attention maps is usually considered a desirable property of ViTs [14, 7, 9]. However, observations in the language domain suggest that self-attention contextualizes tokens, such that the same token has different meanings in different contexts[16]. Contextualization in vision may require a token to receive information not only from same-category tokens, but also from a wider range of different-category tokens such as backgrounds or other objects in the scene. Contextual effects also abound in neuroscience, whereby the responses of neurons and perception are influenced by the context [8, 23, 44, 25, 22, 11, 3, 10]. Therefore, two ideas exist regarding self-attention: a token attends to similar tokens, which could lead to grouping and highlighting the objects; or attends to dissimilar tokens such as backgrounds and different objects, which could lead to stronger contextualization. The former has been supported by many studies, while the latter has been largely ignored in previous studies.

Much like all other deep learning models, though ViTs are successful in many applications, researchers do not have direct access to how information is processed semantically. This issue is particularly important when deploying transformer-based large language models (LLMs) where safety is a priority. As such, there have been studies trying to find feature axes (also known as semantic axes) in the embedding space [18, 5, 6, 13, 34, 19]. A general finding is that embeddings in feedforward layers (i.e. MLP layers) are more semantically interpretable than in self-attention layers [18, 19]. It is believed that the embeddings in the self-attention layers have more superposition, whereas embeddings in the feedforward layers have less superposition due to the expansion of dimensionality [6]. Thus, there has been less focus on finding feature axes in the self-attention layers, and there has been little study addressing interactions between feature axes. In this study, while addressing the role of self-attention, we propose that singular vectors of the query-key interaction are pairs of feature directions. Properties of self-attention heads can be elucidated by studying the properties of their singular modes. We show that those singular vector pairs help semantically explain the interaction between tokens in the self-attention layers.

Our main contributions are as follows:

- We identify a role of self-attention in a variety of ViTs. In many ViTs, especially those with classification training objectives, early layers perform more grouping in which tokens attend more to similar tokens; late layers perform more contextualizing in which tokens attend more to dissimilar tokens. However, this observation has some variability among models and may depend on the training objective: notably, some self-supervised ViTs tend to increase attention to dissimilar tokens in the last few layers.

- We propose a new way to interpret self-attention by analyzing singular modes. Our method goes beyond finding individual feature axes and extends model explainability to the interaction of pairs of feature directions. This approach therefore constitutes enhancing the explainability of transformer models.

In section 2, we state the motivations of this study and list related work. In section 3, we empirically analyze the preference of self-attention between tokens within and between object categories. In section 4, to study the fundamental properties of the query-key interaction, we propose a Singular Value Decomposition method. In section 5, we show that many of the decomposed singular modes are semantic and can be used to interpret the interaction between tokens. In section 6, we discuss the limitations of this study. In section 7, we discuss the main findings and the significance of this study. In the supplementary, we provide an extensive set of visualization examples of the singular modes. Code for this work is available at: https://github.com/schwartz-cnl/DissectingViT.

## 2 Related work

**Attention map properties** The properties of attention maps have been studied since the invention of the ViT. The original ViT paper reported that the model attends to image regions that are semantically meaningful, showing that the $[CLS]$ token (i.e. a special token originally designed as the final hidden vector) attends to objects [14]. Later, a study showed that, in a self-supervised ViT named DINO, the $[CLS]$ attention map has a clearer semantic segmentation property, highlighting the object [7]. Following this idea, studies further showed that the attention map of tokens can highlight parts of an object, and subsequently developed a segmentation algorithm by aggregating attention maps

[31, 39]. Another study on the output of self-attention layers indicates that self-attention may perform perceptual grouping of similar visual objects, rather than highlighting a salient singleton object that stands out from other objects in the image [29]. Most of these studies focus on the $[CLS]$ token attention map or on the outputs of attention maps. Our study, in contrast, seeks to interpret the interactions between tokens within the self-attention layers, to gain insights about properties such as grouping and contextualization.

**Contextualization** Our study is inspired by contextual effects in visual neuroscience, in which neural responses are modulated by the surrounding context [3, 8, 44]. For instance, the response of a cortical visual neuron in a given location of the image is suppressed when the surrounding inputs are inferred statistically similar, but not when the surround is inferred statistically different, thereby highlighting salient stimuli in which the center stands out from the surround [25, 12]. Some of these biological surround contextual effects have been observed in convolutional neural networks [28, 32]. Here our goal is not to address biological neural contextual effects in ViTs, but to dissect contextual interactions in the self-attention layers. It is known that language transformer models have a strong ability to contextualize tokens [16]. However, it's not clear what kinds of contextualization emerge in the ViT. In this study, we seek to understand what kinds of interactions occur between a token and other tokens that carry important contextual information, possibly representing different objects, different parts of an object, or the background.

**Finding feature axes** Finding feature axes is crucial for understanding and controlling model behavior. Since a study found semanticity in the embeddings of feedforward layers in LLMs [18], studies have primarily focused on identifying feature axes in the feedforward layers, and to a lesser extent, in self-attention layers. Similar to the findings in LLMs, a ViT study found that feedforward layers have less mixed concepts and can generate interpretable feature visualizations [19]. Bills et al. proposed a gradient-based optimization method to find explainable directions in LLMs [5]. Later, Bricken et al. proposed a simpler method of sparse autoencoder [6]; though see [21]. These methods have not been extensively applied to ViT studies.

Some studies focused on finding feature directions in the ViTs' self-attention layers. In downstream tasks such as semantic segmentation, researchers empirically found that choosing the key embeddings as features leads to the best performance [37, 2, 1]. A study proposed that the singular value decomposition of the weight matrix is a natural way to find feature directions in any neural network [34]. But they only focused on single feature directions (right singular vectors), and did not consider the feature interaction in the context of self-attention. Another study suggested that singular vectors of value weights and feedforward weights can be used as features in LLMs, but they did not analyze the query-key interaction matrix [30]. Another study in the language domain proposed a singular vector decomposition on the union of the query and key embeddings, but not on the query and key weights [27].

There has been limited work going beyond single features to studying query-key interactions. A study focusing on LLMs proposed that the corresponding columns of query and key matrices are interpretable as pairs [13]. However, this approach does not find features beyond the canonical basis of the query and key embeddings. Another study inferred query-key interactions by jointly visualizing them in a low dimensional space, but their method does not find interacting feature axes [42]. Here, in contrast to previous works, we utilize the singular value decomposition to study the query-key interactions. We propose that left and right singular vectors of the query-key interaction matrix can be seen as pairs of interacting feature directions, and study their properties in ViTs.

## 3 Grouping or contextualizing

Firstly, we empirically study whether an image token (i.e. a patch in the image) attends to tokens belonging to the same objects, different objects, or background. We utilized a dataset that has been applied to studying visual salience [24], namely the Odd-One-Out (O3) dataset [29]. This dataset was also used by Mehrami et al. [29] in their study but they only focused on the output of the attention layers. However, we use a different experimental design that focuses on the attention maps of image tokens. The dataset consists of 2001 images that have a group of similar objects (distractors) and a distinct singleton object (target) (Fig 2 A). Our goal is to examine if the attention map of a

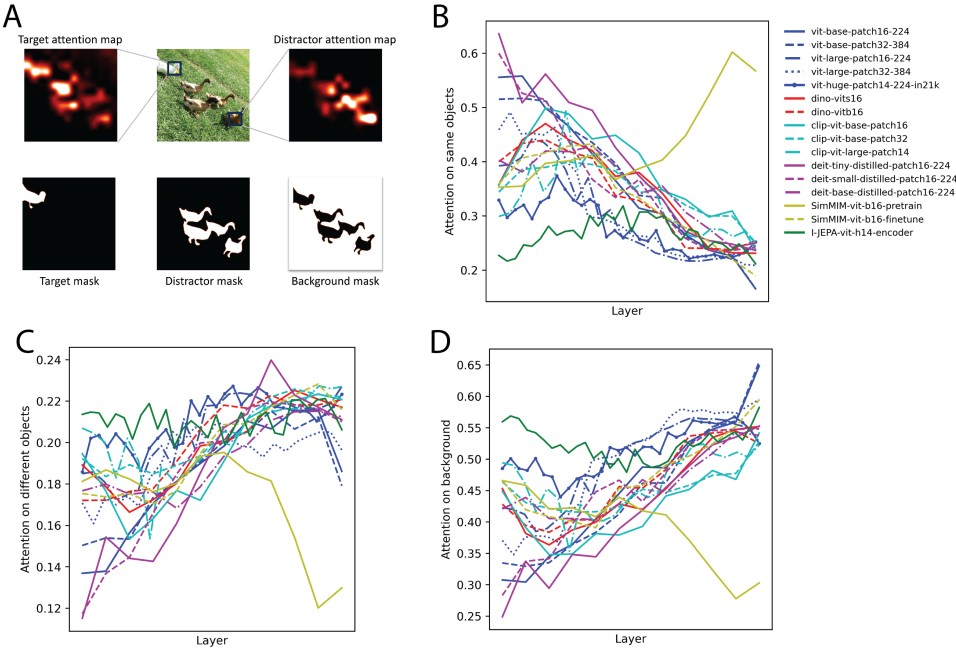

Figure 2: Attention preference in the Odd-One-Out (O3) dataset [24]. A. An example from the O3 dataset. Two tokens are chosen to correspond to the target and distractor in the image. Attention maps using two tokens as queries are computed. We examine the overlap between the attention map of the target, and each of the mask labels of the target, distractor, and background masks. Similarly, we examine the overlap between the attention map of the distractor, and each of the mask labels of the distractor, target, and background. B. Ratio of attention on the same objects (target-target and distractor-distractor attention). The x-axis is normalized layer numbers, from early layers (left) to late layers (right). C. Ratio of attention on the different objects (target-distractor and distractor-target attention). D. Ratio of attention on the background (target-to-background and distractor-background attention)

token of one category (target or distractors) covers more of the same category, different category, or background.

We chose to study 16 different ViT models from 6 families: the original ViT [14], DeiT which uses distillation to learn from a teacher model [38], CLIP which is jointly trained with a text encoder [33], and DINO [7], SimMIM [40], I-JEPA [4] which are self-supervised models with either contrastive or mask prediction loss.

In this study, the "attention score" is defined as the dot product of every query and key pair, which has the shape of the number of tokens by the number of tokens and is defined per attention head. The "attention map" is the softmax of each query's attention score reshaped into a 2D image, which is defined per attention head and token. For each image in the dataset, two tokens are chosen to represent the target and distractor. They are at the location of the maximum value of the down-scaled target or distractor mask. Two attention maps are obtained using the two tokens, each is normalized to sum to 1. Inner products are computed between the two attention maps and three masks, which can be interpreted as the ratio of attention of an object (target or distractor) on the same object, different object, or background. We use target-target, target-distractor, target-background, distractor-target, distractor-distractor, and distractor-background attention to denote the 6 inner products. This measure is computed per layer, head, and image. The averaged measure is shown in Fig 2. Target-target and distractor-distractor attention are categorized as "attention on same objects"; target-distractor and distractor-target attention are categorized as "attention on different objects"; target-to-background and distractor-to-background attention are categorized as "attention on background". The attention on the same objects should be dominant if attention is to perform grouping. We find a trend that in most ViTs the attention on the same objects is dominant in early layers; while there is a trend that in

the deeper layers attention gradually increases on the contextual features such as the background or different objects. However, this observation has some variability among models and may depend on the training objective. For example, the self-supervised SimMIM pre-trained on pixel-level mask prediction shows increased attention on the same objects in later layers. Interestingly, this trend disappears after fine-tuning on a classification task.

This result provides new evidence that self-attention considers contextual features as much or more than similar features in deeper layers. In most ViTs, especially with those with classification training objectives, self-attention prefers the same objects in early layers; in deeper layers, self-attention shifts to contextual information. As far as the authors are aware, this finding has not been reported in previous ViT studies [14, 7, 39, 29].

## 4 Singular value decomposition of query-key interaction

### 4.1 Formulation

In the previous section, we empirically study the allocation of self-attention and find that self-attention does not only do grouping. In this section, we try to find whether this self-attention property can be better understood by analyzing the underlying computation. The self-attention computation is formulated as below, following the convention in the field. Each token is first transformed into three embeddings, namely query, key and value. The output of a self-attention layer is the sum of values weighted by some similarity measures between query and key. The original transformer model used the softmax of the dot-product of the key and query [14]:

$$\text{Attention}(\mathbf{Q}, \mathbf{K}, \mathbf{V}) = \text{softmax}(\frac{\mathbf{Q}^\top \mathbf{K}}{\sqrt{d_k}})\mathbf{V}$$

where $\mathbf{Q}$, $\mathbf{K}$, $\mathbf{V}$ denote the query, key, and value embeddings. They are calculated from linearly transforming the input sequence $\mathbf{X} = \{\mathbf{x}_1, ..., \mathbf{x}_L\} \in \mathbb{R}^{d \times L}$, where $d$ is the input embedding size, L is the sequence length,

$$\mathbf{Q} = \mathbf{W}_q \mathbf{X} \in \mathbb{R}^{d_k \times L}$$
$$\mathbf{K} = \mathbf{W}_k \mathbf{X} \in \mathbb{R}^{d_k \times L}$$
$$\mathbf{V} = \mathbf{W}_v \mathbf{X} \in \mathbb{R}^{d_v \times L}$$

where $\mathbf{W}_q \in \mathbb{R}^{d_k \times d}$, $\mathbf{W}_k \in \mathbb{R}^{d_k \times d}$, $\mathbf{W}_v \in \mathbb{R}^{d_v \times d}$ are trainable linear transformations that transform the input embedding to the key, query, and value space. Sometimes a bias term is also added to the transformation. Since the bias term does not depend on the input embedding, we do not include it in our analysis of token interactions. In the formula of the attention output, the part that contains the query and key interaction is named the attention score. In this case which is based on the dot-product, the attention score between two tokens $\mathbf{x}_i$ (query) and $\mathbf{x}_j$ (key) is

$$a_{ij} = \mathbf{q}_i^\top \mathbf{k}_j = \mathbf{x}_i^\top \mathbf{W}_q^\top \mathbf{W}_k \mathbf{x}_j$$

The attention score solely depends on the combined matrix $\mathbf{W}_q^\top \mathbf{W}_k$ as a whole [15], which represents the query-key interaction. To better understand the behavior of this bilinear form, we factor the matrix using the singular value decomposition,

$$\mathbf{W}_q^\top \mathbf{W}_k = \mathbf{U}\mathbf{\Sigma}\mathbf{V}^\top$$

where $\mathbf{U} = \{\mathbf{u}_1, ..., \mathbf{u}_{d_k}\} \in \mathbb{R}^{d \times d_k}$ is the left singular matrix composed of left singular vectors, $\mathbf{V} = \{\mathbf{v}_1, ..., \mathbf{v}_{d_k}\} \in \mathbb{R}^{d \times d_k}$ is the right singular matrix composed of right singular vectors, $\mathbf{\Sigma} = \text{diag}(\sigma_1, ..., \sigma_{d_k}) \in \mathbb{R}^{d_k \times d_k}$ is a diagonal matrix composed of singular values. We will refer to *the nth singular mode* as the set $\{\mathbf{u}_n, \sigma_n, \mathbf{v}_n\}$. Then the attention score between two tokens can be decomposed into singular modes.

$$\mathbf{x}_i^\top \mathbf{W}_q^\top \mathbf{W}_k \mathbf{x}_j = \sum_{n=1}^{d_k} \mathbf{x}_i^\top \mathbf{u}_n \sigma_n \mathbf{v}_n^\top \mathbf{x}_j$$

Consider the input embeddings projected onto the left and right singular vectors, i.e. $\mathbf{x}^\top \mathbf{u}_n$ and $\mathbf{x}^\top \mathbf{v}_n$. The attention score is non-zero when the two embeddings have a non-zero dot-product with the

corresponding left and right singular vectors within the same singular mode. In other words, if one embedding happens to be in the direction of a left singular vector, it only attends to tokens that have a component of the corresponding right singular vector. It can be thought of as a left singular vector "query" looking for its right singular vector "key".

## 4.2 Similarity between left and right singular vectors

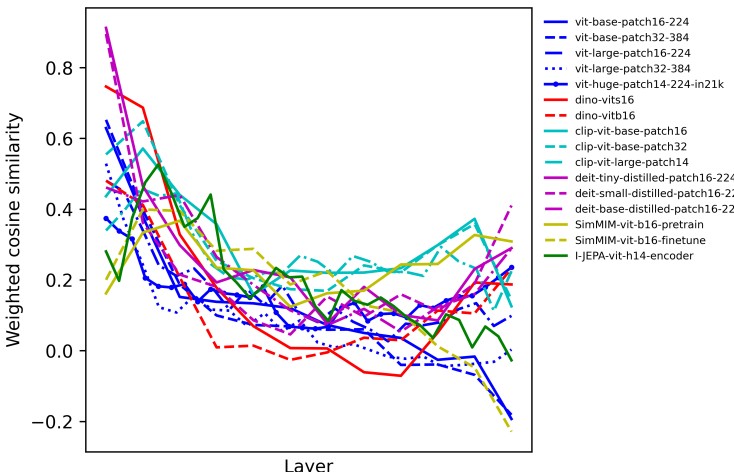

Figure 3: Cosine similarity between left and right singular vectors. The cosine similarity is computed per head and singular mode. The weighted average value of cosine similarity is computed with weights of corresponding singular values.

To determine if self-attention performs grouping or combines contextual information, we examine whether tokens in different layers have higher attention scores with similar tokens or dissimilar tokens. This can be measured for each singular mode by how much the left singular vector is aligned with the right singular vector, more specifically, the cosine similarity between the left singular vector and the right singular vector. A high cosine similarity value means tokens attend to similar tokens (to itself if the value is 1); a low value means tokens attend to dissimilar tokens (to orthogonal tokens if 0; to opposite tokens if negative). The average cosine similarity is weighted by the singular values with the assumption that singular modes with higher singular values are more influential to the total attention score $\cos_{avg} = \sum_i \frac{\sigma_i}{\sum \sigma_j} \cos_i$. We find that the averaged cosine similarity is high in early layers, and there is a decreasing trend in deeper layers (Fig 3). In some models, the averaged cosine similarity drops to 0 in some middle layers. The cosine similarity distribution and singular value spectrum of the vit-base-patch16-224 model is provided in the Supplementary Figures S1 and S2.

Though we find a general trend that attention changes from attending more to the similar tokens to dissimilar tokens from early layers to late layers, some ViTs have a more complex trend that increases attention to similar tokens in the last few layers (Fig 3). Models that have this "concave" trend are SimMIM-vit-b16-pretrain, Dino models, Deit models, and huge ViT models. Most of them either have self-supervised objectives or distillation regularizations. We hypothesize that the last layers may behave differently because they are closer to the training target, and so the training objective may have more influence. We think that self-supervised objectives, such as reconstructing masked patches, require stronger consistency between tokens, and thus more attention is allocated to similar tokens in the higher layers; while the classification objective requires gathering information from different aspects of a scene, and thus more attention is allocated to dissimilar tokens. This hypothesis is supported by the cosine similarity plot (Fig 3) of the SimMIM models, which shows in the last few layers of the pre-trained model increased attention to similar features. This matches the observation in the literature, that the SimMIM model has more local attention [41]. However, we find that the SimMIM model fine-tuned on ImageNet classification has a trend of decreased attention to similar features, similar to most of the classification models. Although I-JEPA is trained with a self-supervised objective predicting latent representations, the cosine similarity for the I-JEPA

encoder does not show increased attention to similar tokens in the last few layers. The I-JEPA model is known to have excellent linear-probing performance, and thus we think it may behave more similarly to a classification model. The self-supervised objective of I-JEPA may be more apparent in the I-JEPA predictor (also a transformer). When we run the cosine similarity analysis on the predictor module instead of the encoder, we find that the cosine similarity is overall high (Supplementary FigureS3). The role of the training objective on internal model behavior is an interesting topic for future research.

It is known that embeddings in transformer models are to some extent anisotropic [17, 26, 20], which means the expected value of cosine similarity of two random sampled inputs tends to be positive. We indeed find anisotropy effects in all the models we examined using cosine similarity (Supplementary Figure S4) (though see other metrics [35]). If we treat anisotropy level as a baseline for cosine similarity, the effect shown in Fig 3 still exists but the self-attention is less biased to similar tokens (Supplementary Figure S4).

There is a further implication of the singular value decomposition approach. The left and right singular vectors of each attention head are two incomplete orthonormal bases of embedding. We suggest that these bases are feature directions since they are intrinsic properties of the self-attention layer. The query and key embeddings can be made arbitrary, since one can change the basis without affecting the attention score. However, the singular vectors are invariant to the change of basis. If an invertible matrix $\mathbf{A} \in \mathbb{R}^{d_k \times d_k}$ acts on the query and key weights as $\mathbf{W}_q \rightarrow \mathbf{A}^\top \mathbf{W}_q$ and $\mathbf{W}_k \rightarrow \mathbf{A}^{-1}\mathbf{W}_k$, then the attention score does not change but the query and key embeddings change. The singular vector decomposition of $(\mathbf{A}^\top \mathbf{W}_q)^\top \mathbf{A}^{-1}\mathbf{W}_k$ stays the same as decomposing $\mathbf{W}_q^\top \mathbf{W}_k$. Thus singular vectors are uniquely special and may show interesting properties. Due to the sign ambiguity of the singular value decomposition, we consider the opposite directions of singular vectors also as feature directions.

## 5    Semanticity of singular modes

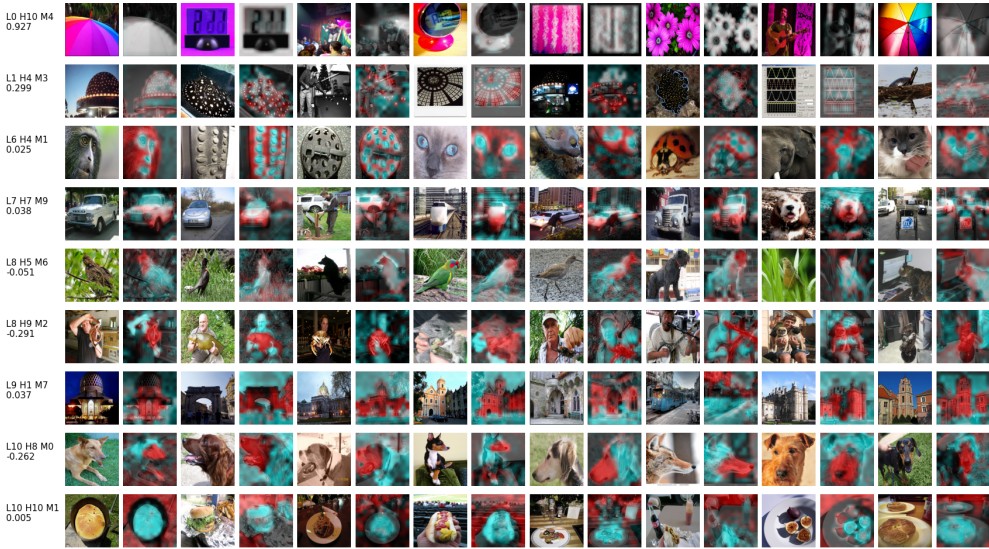

Figure 4: Examples of optimal attention images of singular modes and query and key map in dino-vitb16. Optimal attention images are found from the Imagenet validation set that induce the largest attention score (sorted by the product of the maximum of query map and maximum of key map). The red and cyan channels are the projection values of embedding onto the left and right singular vectors of a singular mode. They correspond to query and key. The white area is where the query map and key map overlap. The name code we assign to singular modes specifies the layer, head, and mode numbers. For example, "L1 H4 M3" means layer 1, head 4, and mode 3. The value below indicates the cosine similarity between the left and right singular vectors.

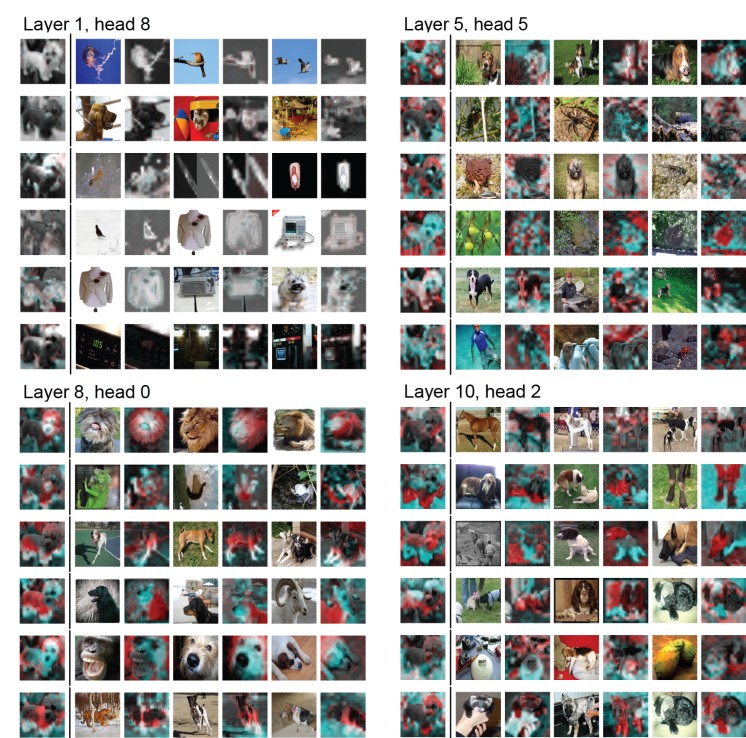

Figure 5: Visualization of a single image with multiple modes. We pick an example dog image from the ImageNet dataset and use the dino-vitb16 model. Top 6 modes (ordered by the contribution to the attention score) for example layers and heads are shown. See Supplementary Figure S17 for extended mode visualizations of this image.

The singular value decomposition of self-attention offers an intuitive way to explain the self-attention layer. A feature represented by a left singular vector attends to the feature represented by the corresponding right singular vector. The feature of a singular vector can be found by finding the image that has the maximum embedding projection on the singular vector. Similarly, the typical interactions of a singular mode can be identified by finding the image that has the maximum product of the projections on a singular vector pair. Previous studies on the explainability of deep learning models only focused on the explainability of single neurons or individual feature axes. The singular value decomposition extends model explainability to the interaction of pairs of "neurons" (i.e. singular vectors). Note that this is very different from the standard approach of visualizing the attention map of the $[CLS]$ token without addressing interactions between tokens [14, 7, 31].

Some example modes from dino-vitb16 are shown in Fig. 4. For each mode, we show the top 8 images in the Imagenet (Hugging Face version) [36] validation set that induce the largest attention score. For each image, a query map (red channel in the figure) and a key map (cyan channel in the figure) are obtained by projecting the embedding onto the left and right singular vectors. The embedding is obtained from the input of the self-attention layer, i.e. the output of the layer normalization. Each map tells what information the left or right singular vector represents. Jointly, the highlighted regions in the query map attend to the highlighted regions in the key map. In other words, the information in the highlighted regions of the key map flows to the highlighted regions of the query map. More examples are shown for a range of ViT architectures in the Supplementary Figures S5 - S16.

In early layers, singular vectors usually represent low-level visual features like color or texture, and sometimes positional encoding. In higher layers, singular vectors can represent more complex visual features like parts of objects or whole objects. As shown in the previous sections, high attention scores can be induced between similar tokens (more often in early layers) or dissimilar tokens (more often in late layers). The correspondence to image structure for similar and dissimilar tokens can be seen in the query and key maps. For the modes with high cosine similarity, query and key maps are

similar which could represent color, texture, parts, objects, or positional encoding. For the modes with low cosine similarity, query and key maps look different which could represent different object parts, different objects, or foreground and background. Some examples include: in "L6 H4 M1" the animal face (query) attends to eyes, nose and mouth (key); in "L7 H7 M9" the lower part of a car attends to the upper part of a car and wheels; in "L8 H9 M2" the fish or other things in hand attend to human; in "L10 H10 M1" the background attends to the food.

To show the hierarchical information process across layers, we show an example dog image and example attention heads along with optimal images for top modes in Fig. 5. The late layers capture more semantic information such as the parts of a dog or animal, and a hand with an animal. The early layers capture low-level properties like color. We show more examples in Supplementary Figure S17-S19.

The attention between dissimilar tokens could be thought of as providing contextual information to a given token. In the part-to-part case, finding more parts of an object increases the confidence of finding the object and helps merge smaller concepts into a larger concept. In the object-to-object case, an object attending to a different object could add additional attributes to it, for example, a fish attending a human may add the attribute "be held" to the fish tokens, which helps understanding of the whole scene. These interactions between tokens, though conceptually simple, as far as the authors are aware, have not been reported before this study. This result further supports the idea that self-attention combines contextual information from dissimilar tokens such as backgrounds or different objects.

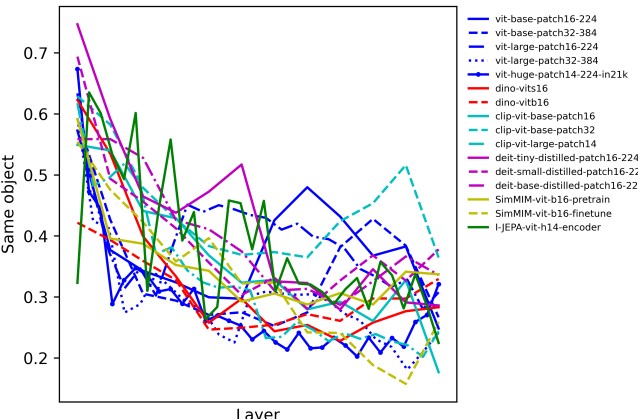

Figure 6: The probability that the left and right singular vectors highlight the same object in maximum attention images.

Finally, we study whether tokens prefer to attend to the same object or different objects at the singular mode level. We choose to use a semantic segmentation dataset, namely ADE20K [43]. We first find the top 5 images that induce maximum attention of a singular mode, then find the optimal objects in each image that have the maximum projections on the left and right singular vectors per object area. The probability of the left and right singular vectors having the same optimal object is computed with the weight of singular values, following the same method in the previous experiment. We find that, in early layers, there is a higher probability that the left and right singular vectors attend to the same object; in late layers, the probability is lower, though the variability between models is considerably large (Fig. 6). This result further supports that self-attention performs more grouping in early layers; in late layers, tokens attend to different objects which could contextualize the token with background information.

## 6   Limitation

We are aware of some limitations of this study and interesting open questions that remain. There is behavioral variability between the models, which may be due to the distinct training objectives. Identifying how the training paradigm alters the learned embedding space is a potential future

direction to explore. We have focused on the query-key interactions in the self-attention, and future studies could address the role of the value matrix.

## 7 Discussion

Inspired by the observation that self-attention gathers information from relevant tokens within an object, and the importance of contextualization in neuroscience, we study fundamental properties of token interaction inside self-attention layers in ViTs. Both empirical analysis of the Odd-One-Out (O3) dataset, and singular decomposition analysis of singular modes for the Imagenet dataset, show that in early layers the attention score is higher between similar tokens, while in late layers the attention score is higher between dissimilar tokens.

The singular decomposition analysis provides a new perspective on the explainability of ViTs. Two directions (left and right singular vectors) in the embedding space could be analyzed in pairs to interpret the interaction between tokens. Using this method, we find interesting semantic interactions such as part-to-part attention, object-to-object attention, and foreground-to-background attention which have not been reported in previous studies. Our reported findings provide evidence that self-attention in vision transformers is not only about gathering information between tokens with similar embeddings, but a variety of interactions between a token and its context. The method of analyzing singular vectors can be easily adapted to study token interactions in transformer networks trained on other modalities like language. Adapting this method to real-world applications can increase transparency of what the transformer models are capturing.

## Acknowledgements

A.P. was supported by the Research Experiences for Undergraduates (REU) Site Scientific Computing for Structure in Big or Complex Datasets, NSF grant CNS-1949972. O.S. was funded by the University of Miami Provost Research Award.

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

# A Supplemental material

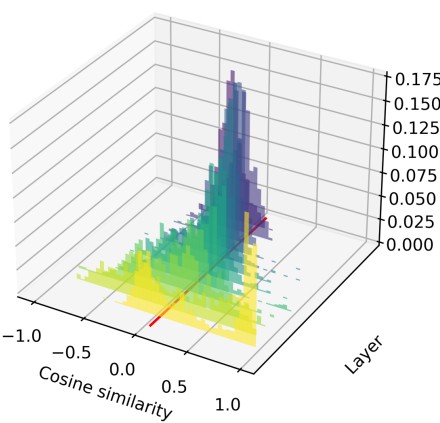

google/vit-base-patch16-224

Figure S1: Histogram of cosine similarity between the left and right singular vector in ViT-base-patch16-224. The yellow layers are earlier layers; the blue layers are later layers. The red line indicates 95% confidence interval, which is calculated from embeddings sampled from a random distribution.

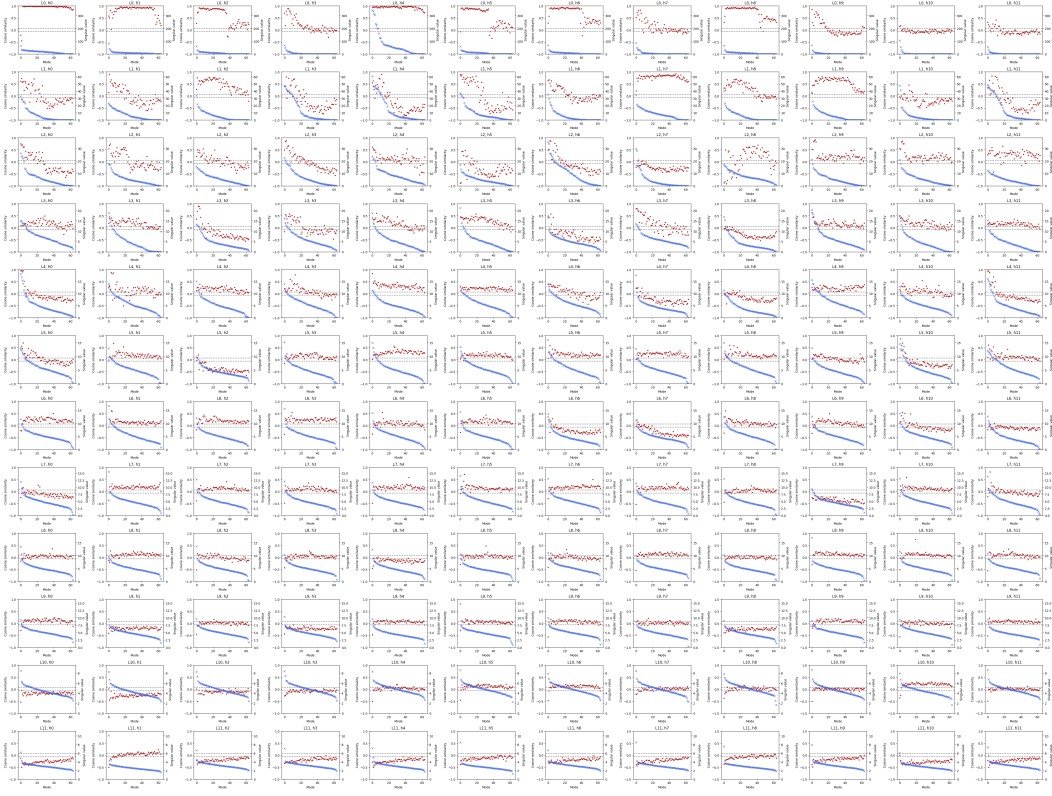

Figure S2: Singular value spectrum (blue) and cosine similarity (red) in ViT-base-patch16-224. Row number indicates layer number. Column number indicates head number. The dotted line indicates 95% confidence interval, which is calculated from embeddings sampled from a random distribution.

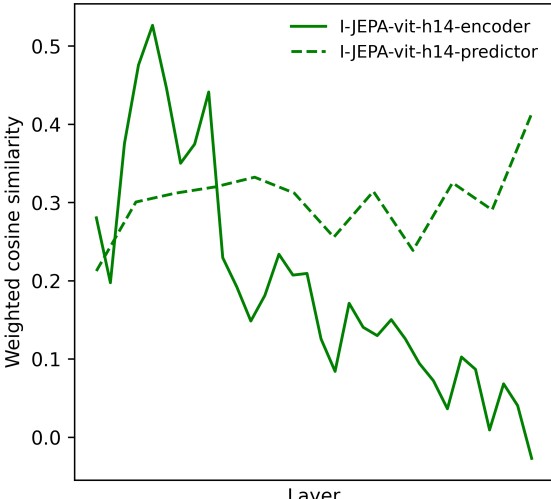

Figure S3: Cosine similarity between left and right singular vectors of the I-JEPA encoder and predictor modules.

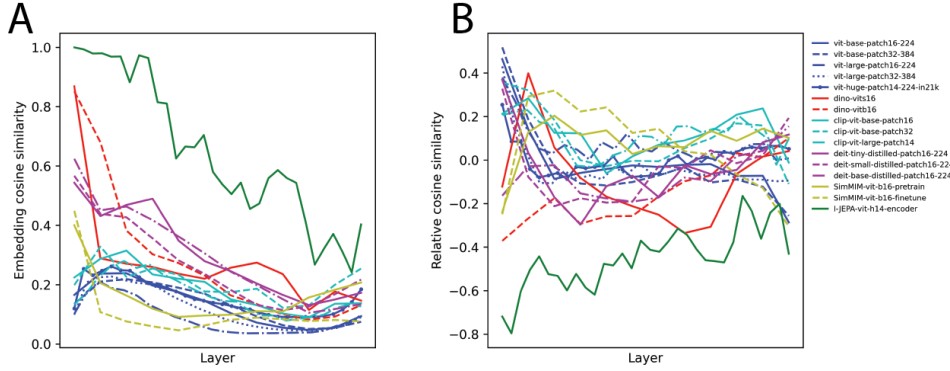

Figure S4: Anisotropy effects in ViTs. A. Averaged embedding cosine similarity between the center tokens of different images from the Imagenet validation set. Consisting with previous studies, the cosine similarities are all positive, which is referred to as anisotropy or cone effect. B. Considering A as the baseline, relative cosine similarity is defined as subtracting cosine similarity between left and right singular vectors by the embedding cosine similarity in A.

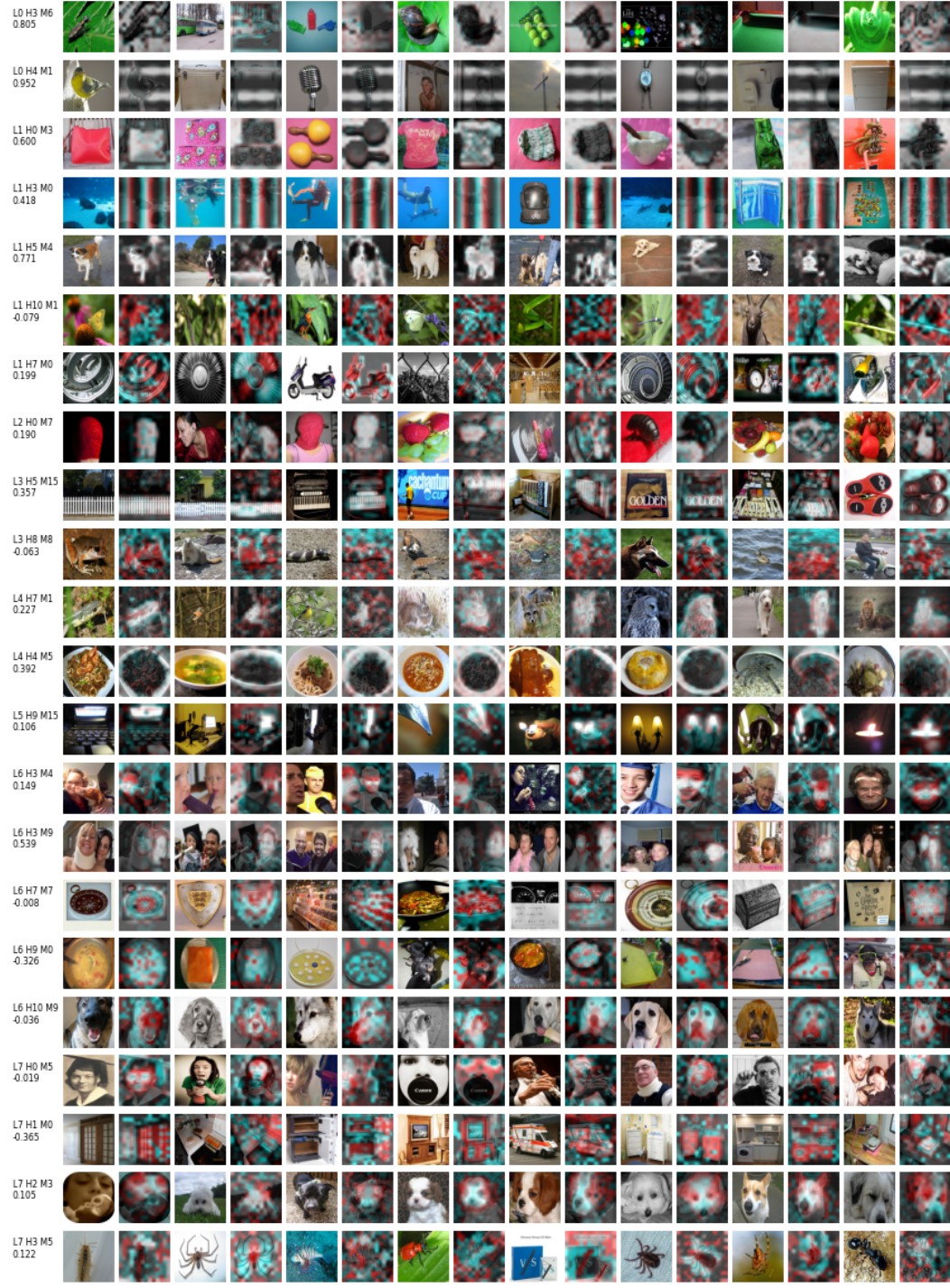

Figure S5: Examples of semantic singular modes in ViT-base-patch16-224 (part 1).

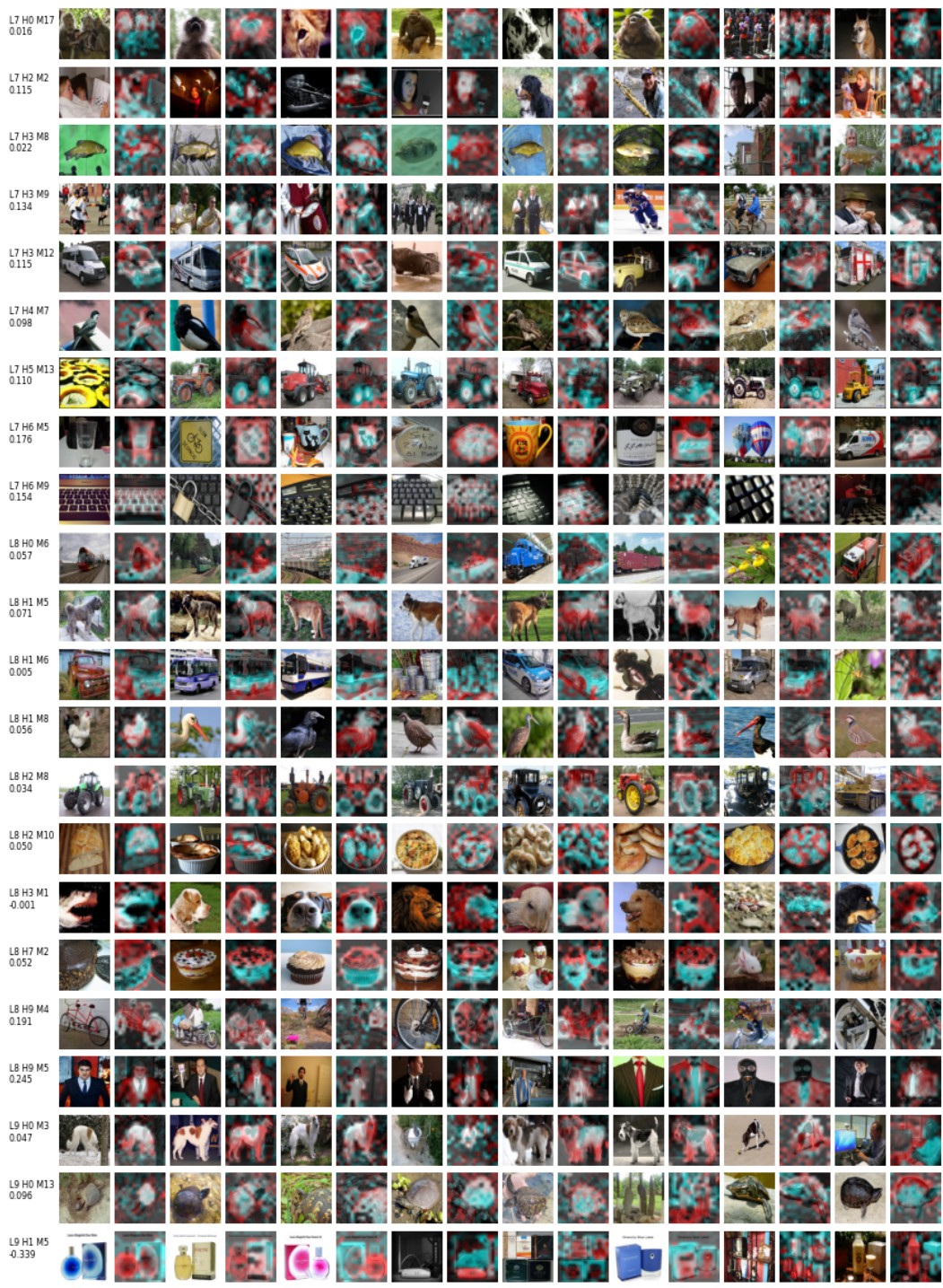

Figure S6: Examples of semantic singular modes in ViT-base-patch16-224 (part 2).

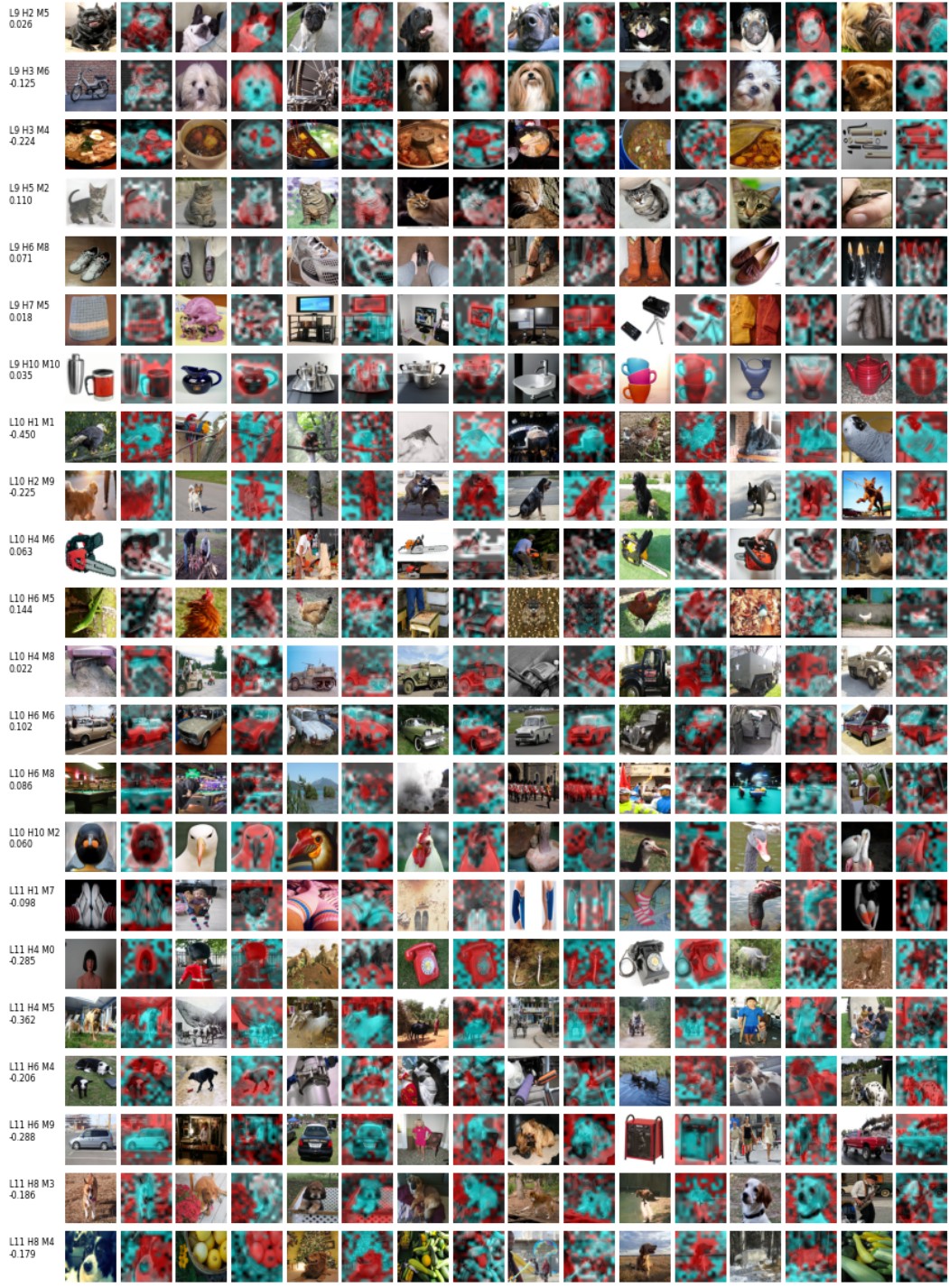

Figure S7: Examples of semantic singular modes in ViT-base-patch16-224 (part 3).

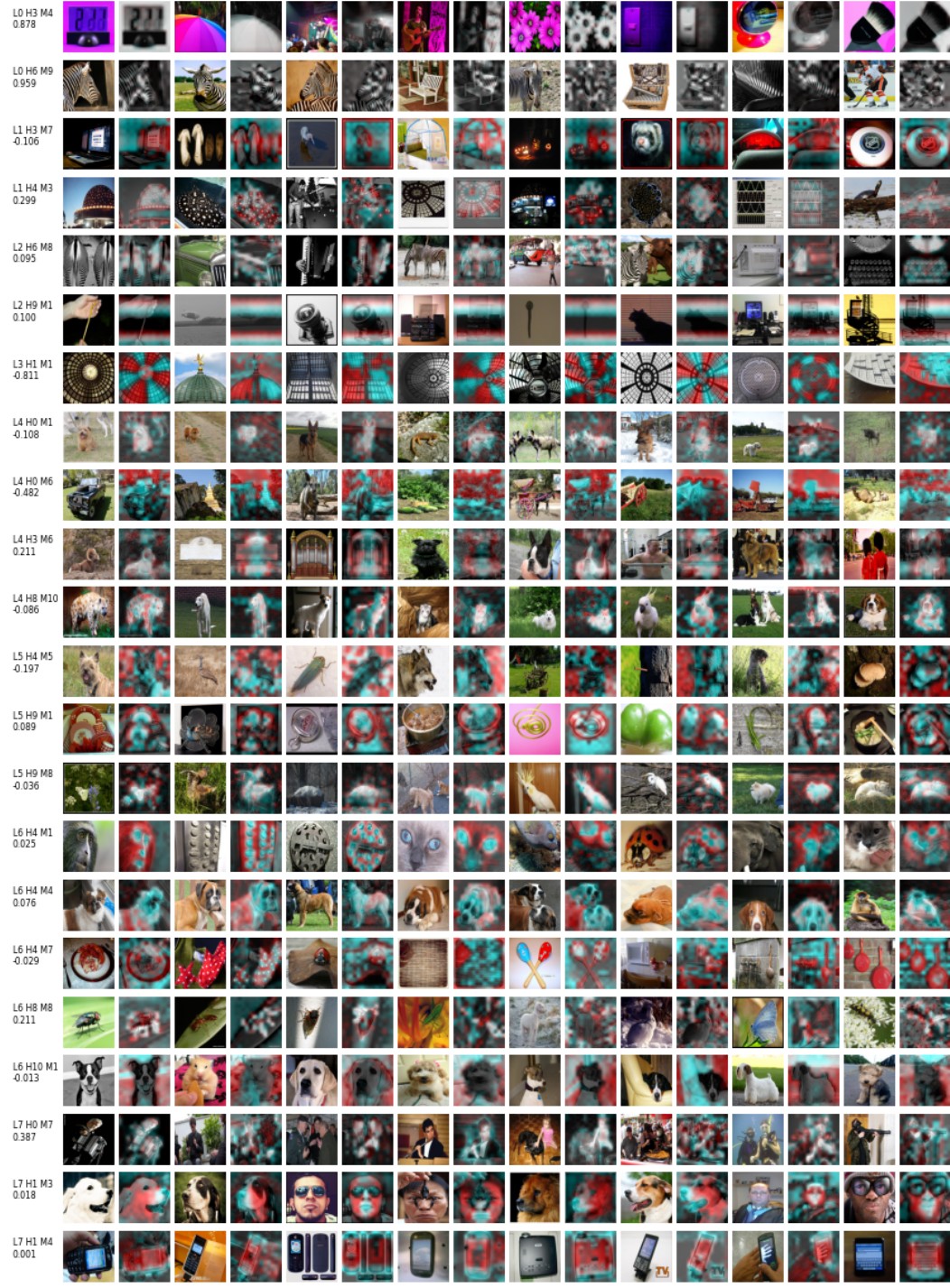

Figure S8: Examples of semantic singular modes in dino-vitb16 (part 1).

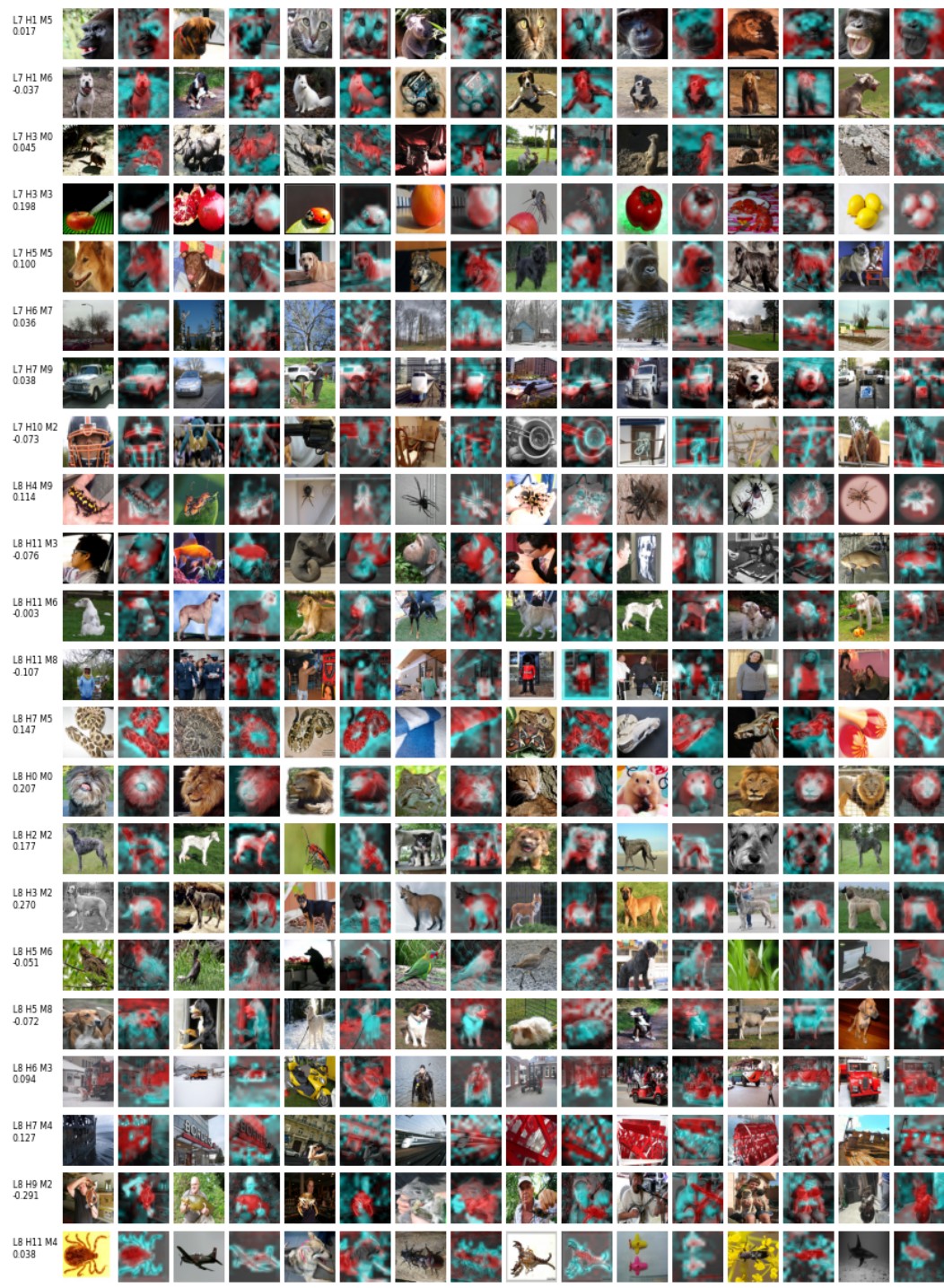

Figure S9: Examples of semantic singular modes in dino-vitb16 (part 2).

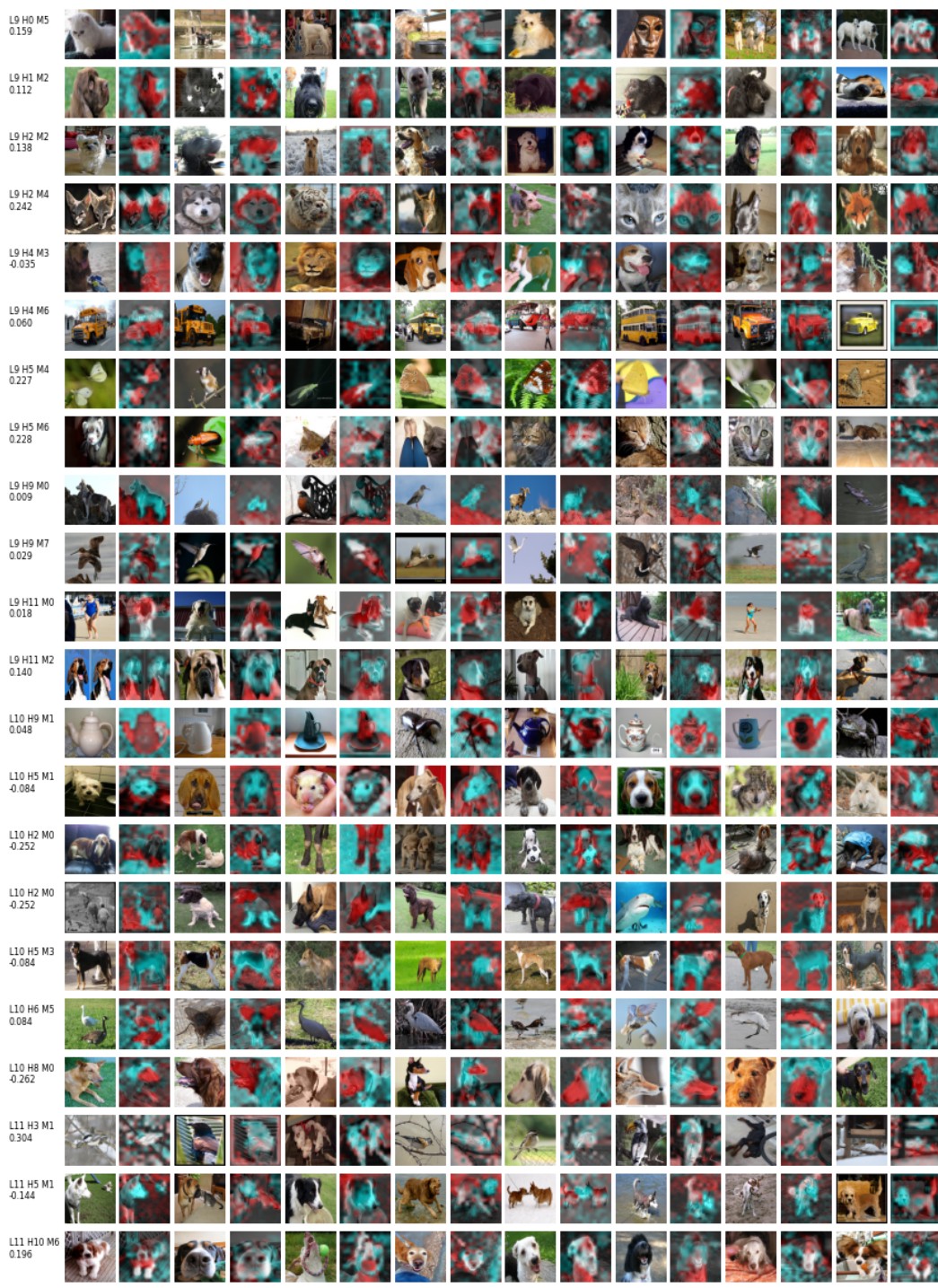

Figure S10: Examples of semantic singular modes in dino-vitb16 (part 3).

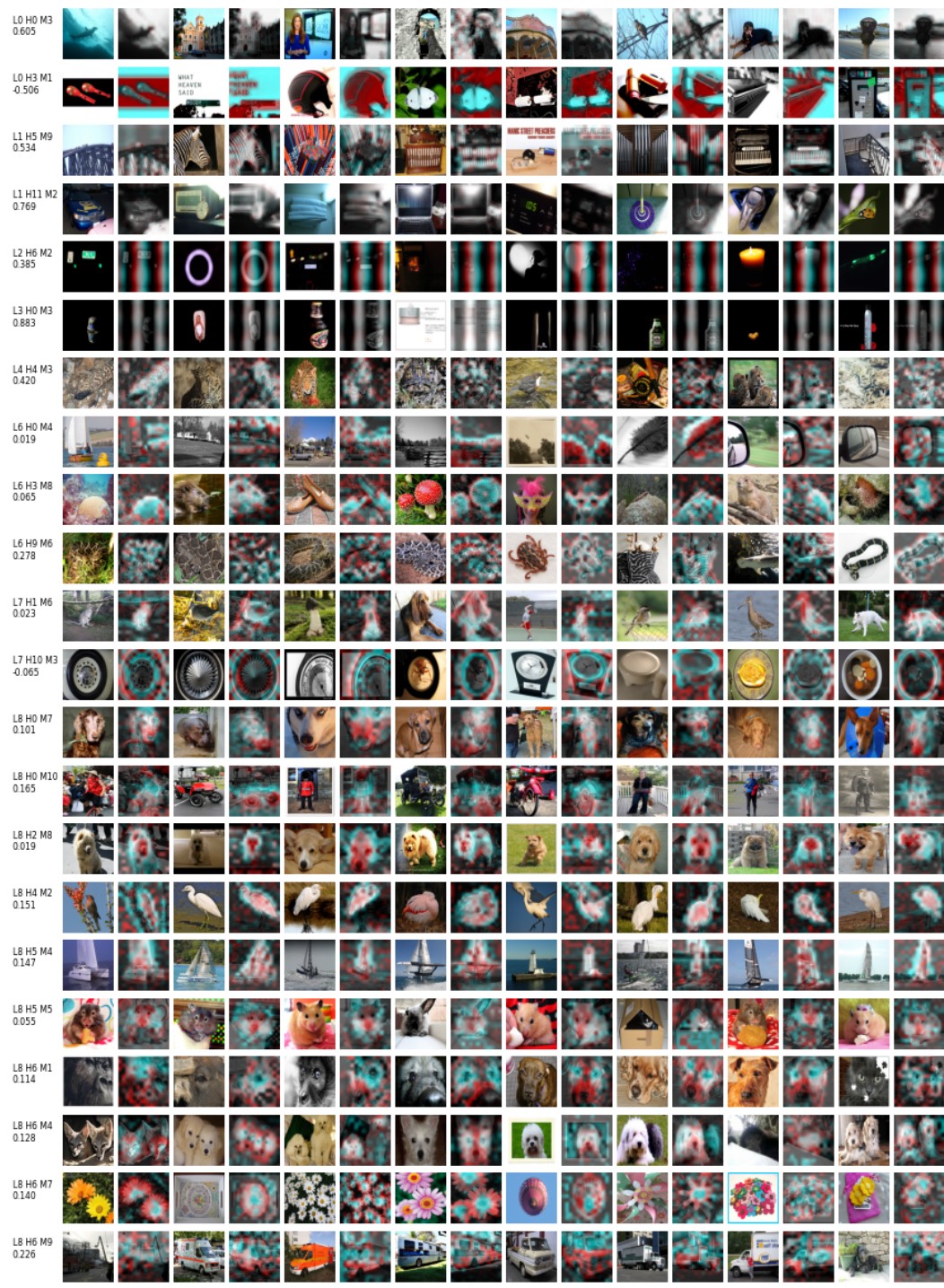

Figure S11: Examples of semantic singular modes in deit-base-distilled-patch16-224 (part 1).

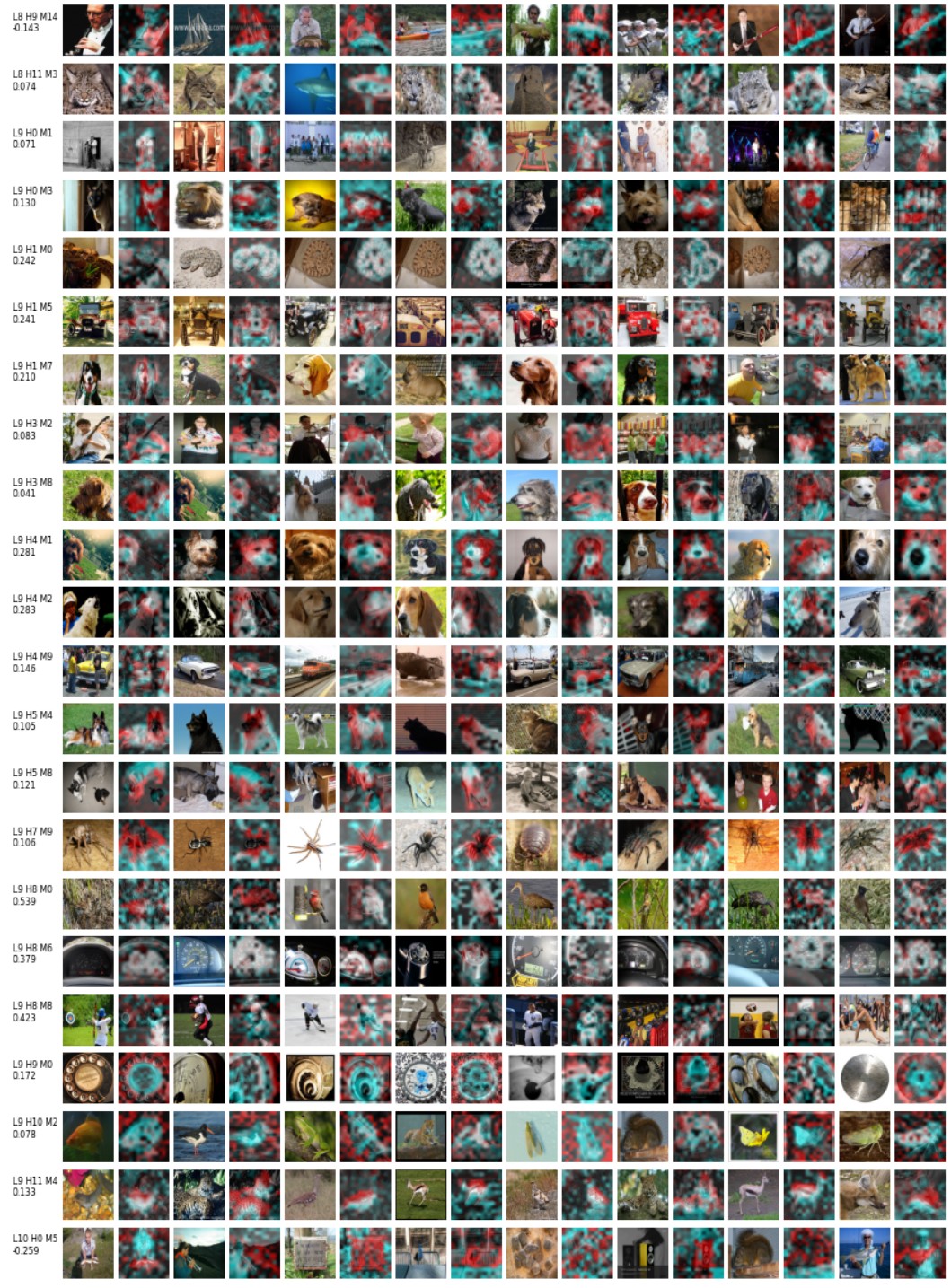

Figure S12: Examples of semantic singular modes in deit-base-distilled-patch16-224 (part 2).

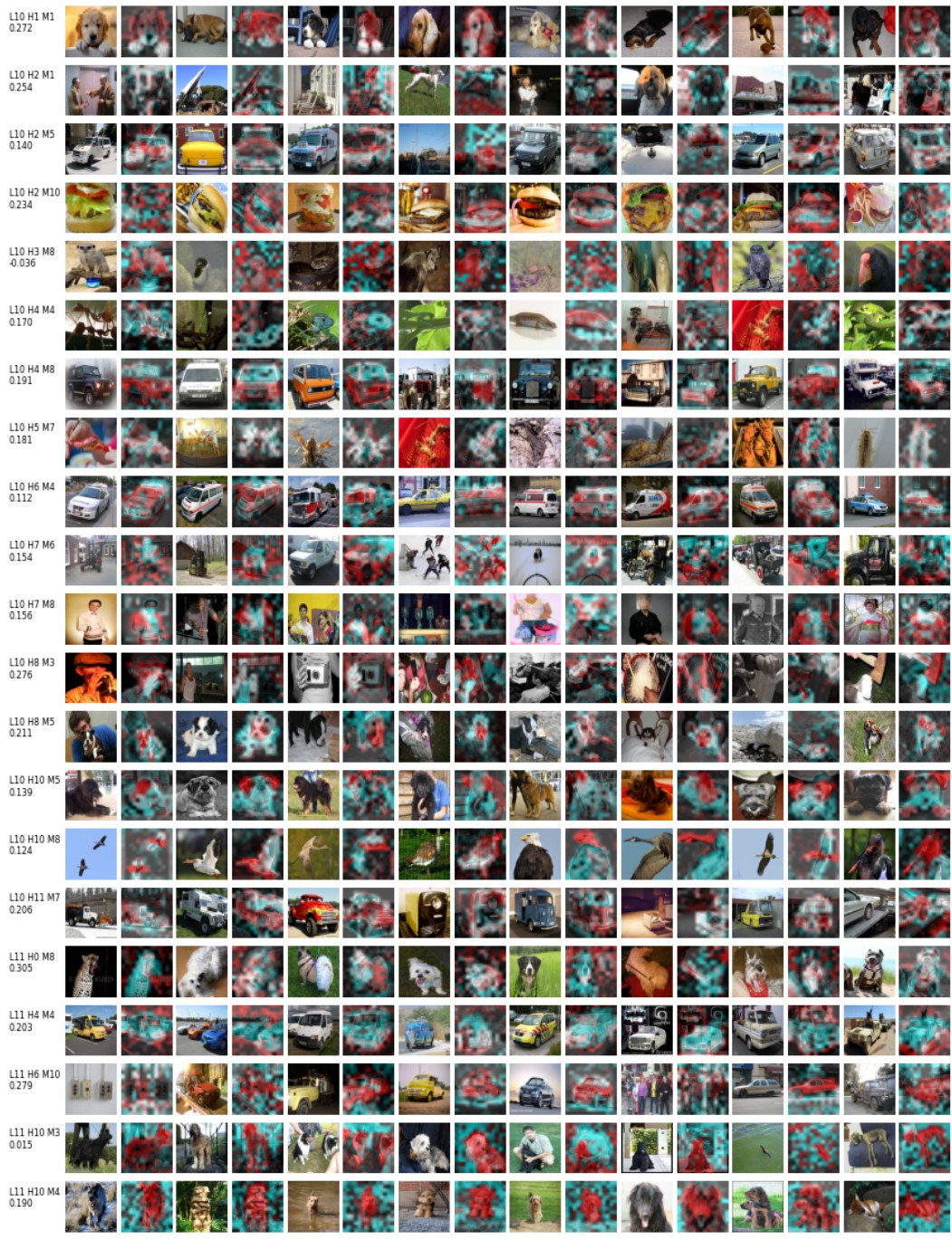

Figure S13: Examples of semantic singular modes in deit-base-distilled-patch16-224 (part 3).

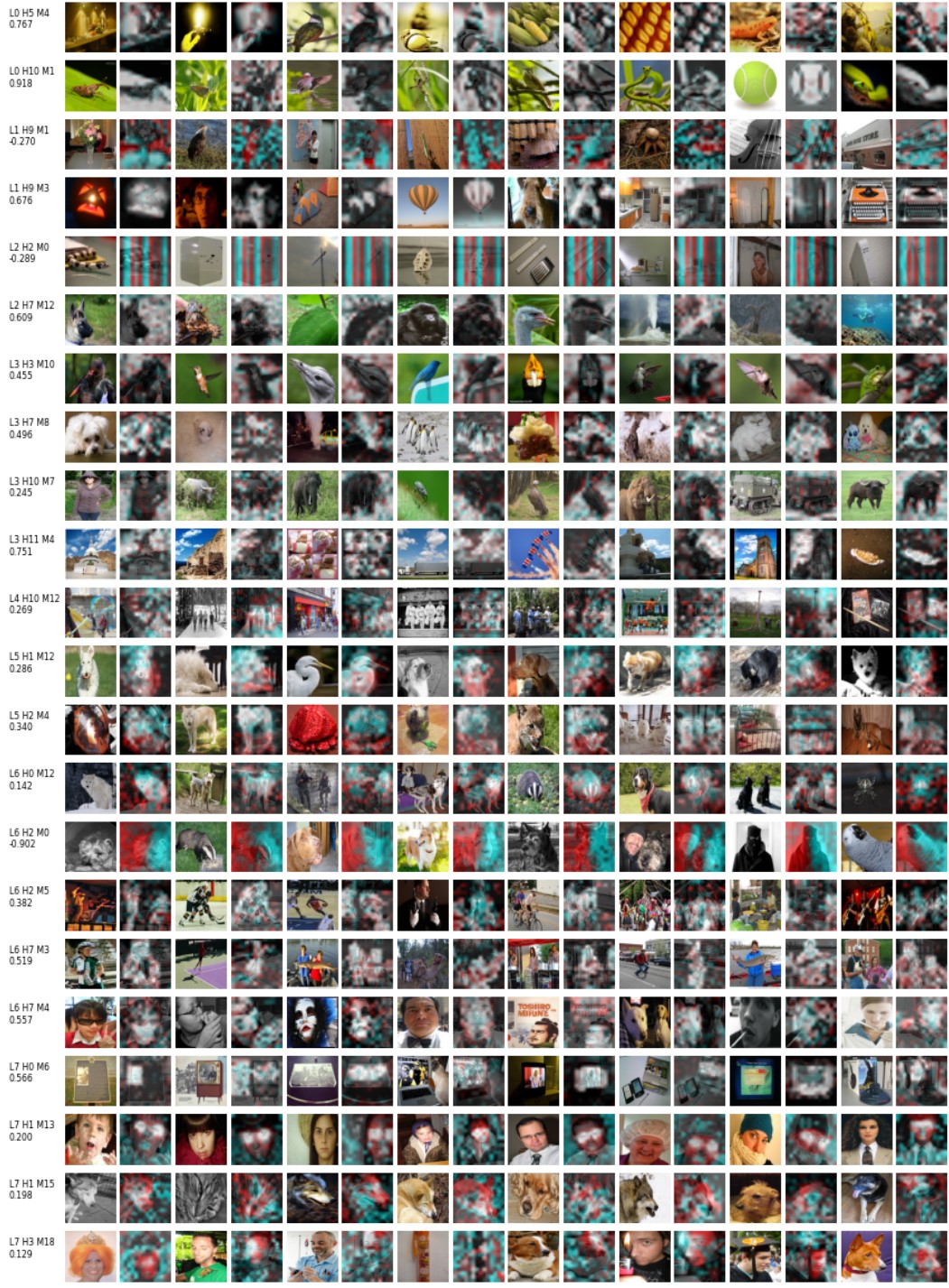

Figure S14: Examples of semantic singular modes in clip-vit-base-patch16 (part 1).

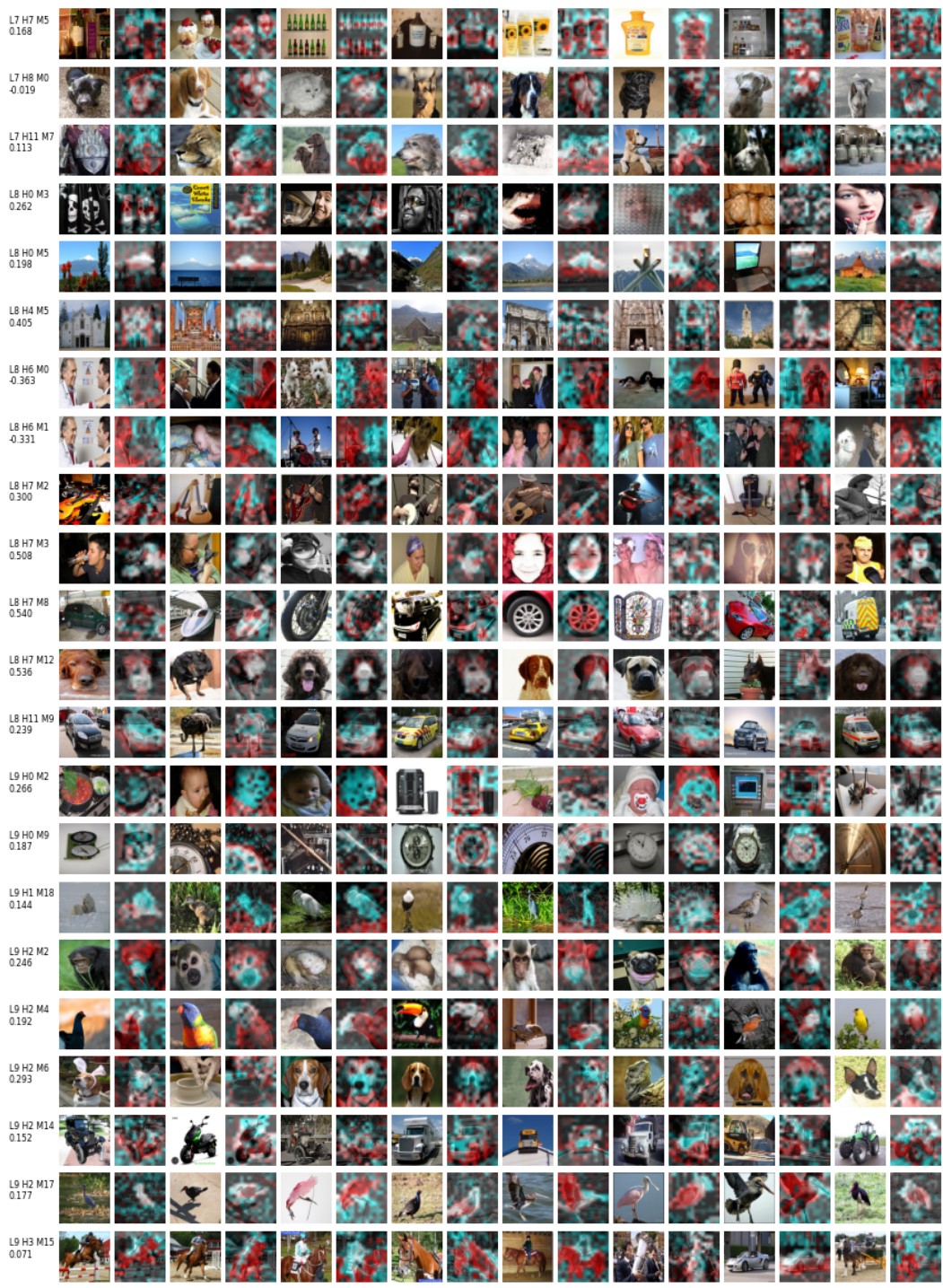

Figure S15: Examples of semantic singular modes in clip-vit-base-patch16 (part 2).

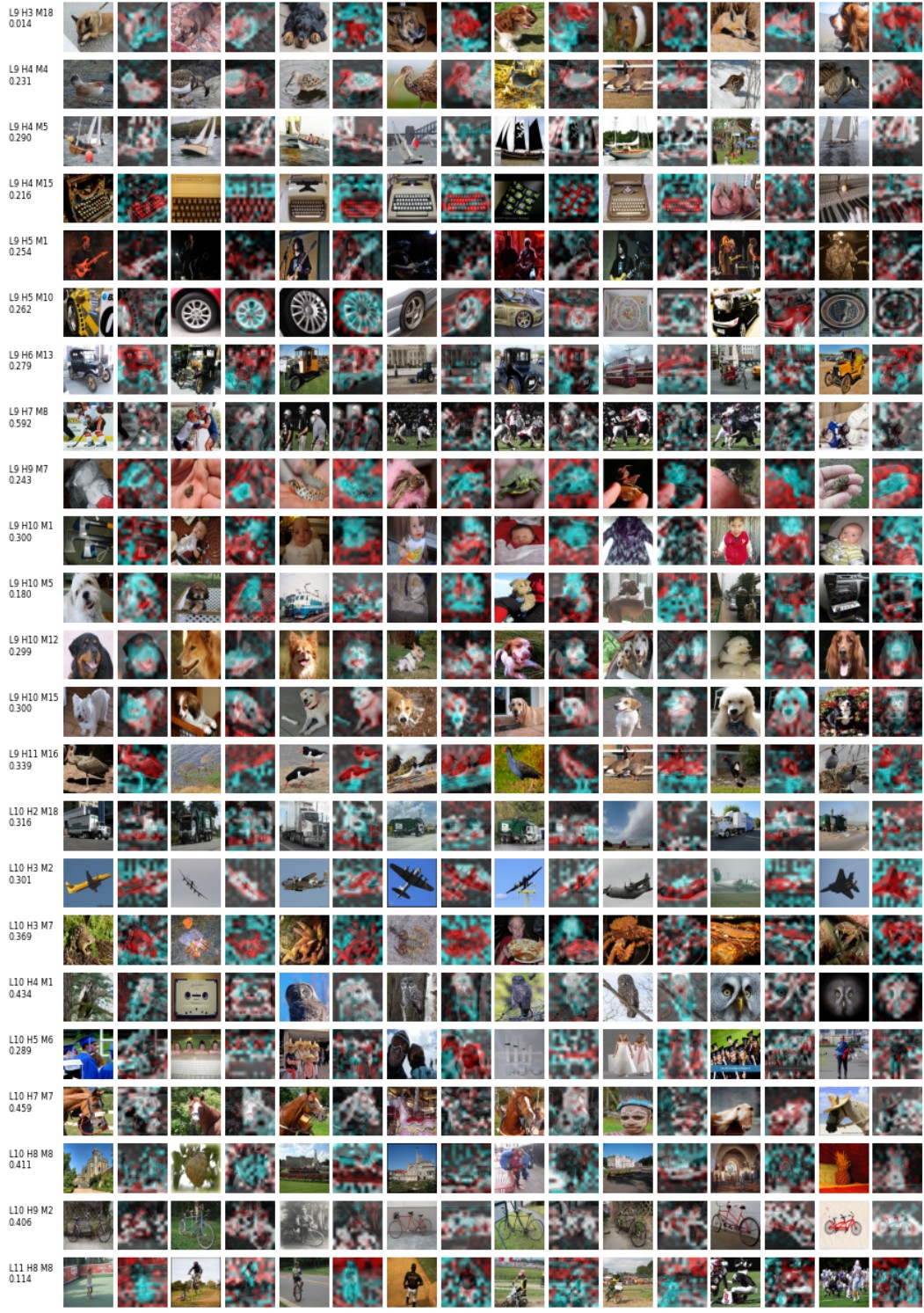

Figure S16: Examples of semantic singular modes in clip-vit-base-patch16 (part 3).

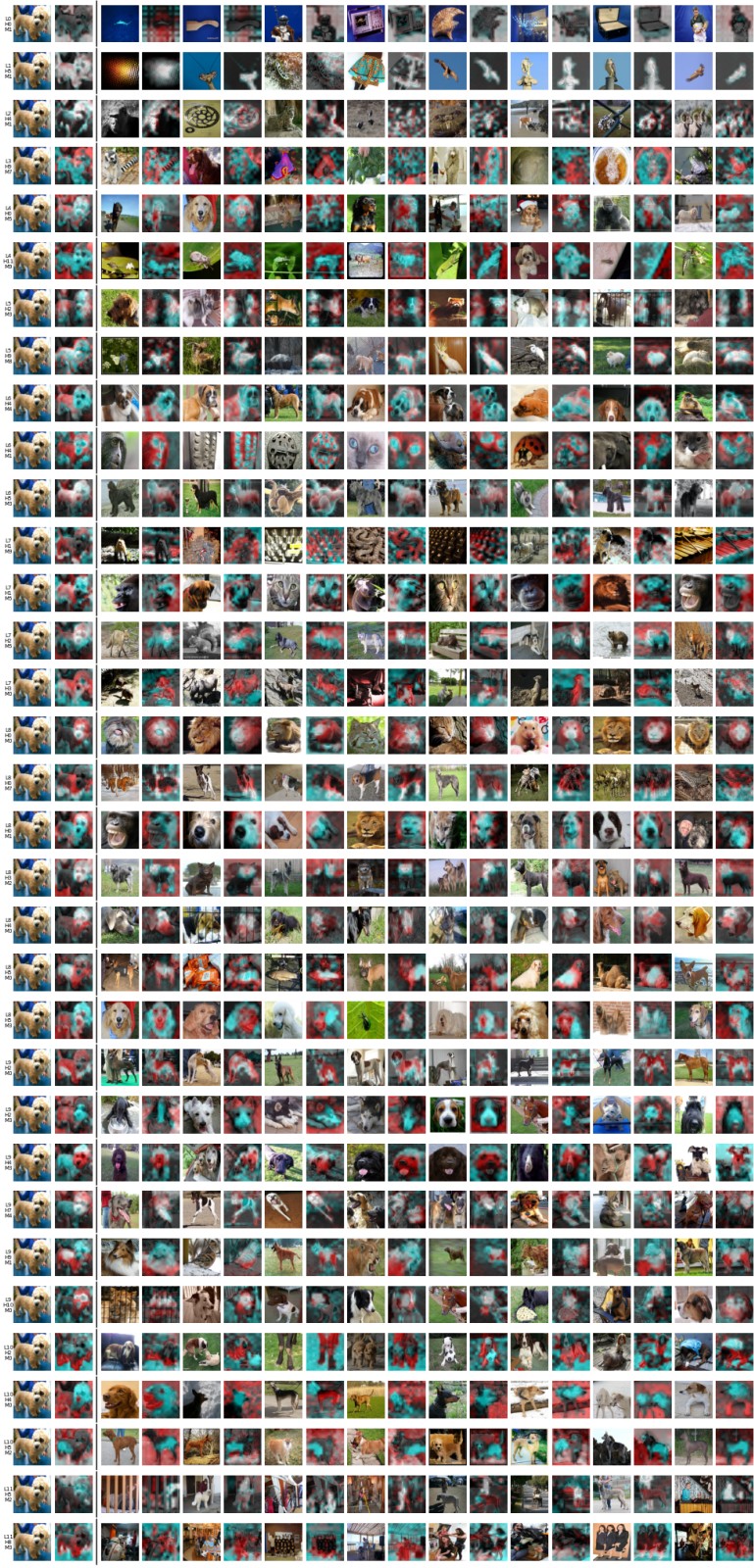

Figure S17: Singular mode maps of a dog image in dino-vitb16. We hand-pick modes to show the variety of information interactions within this image. The left two columns are the original image and corresponding singular mode maps. Other columns are the top 8 images that induce the highest attention through the corresponding mode.

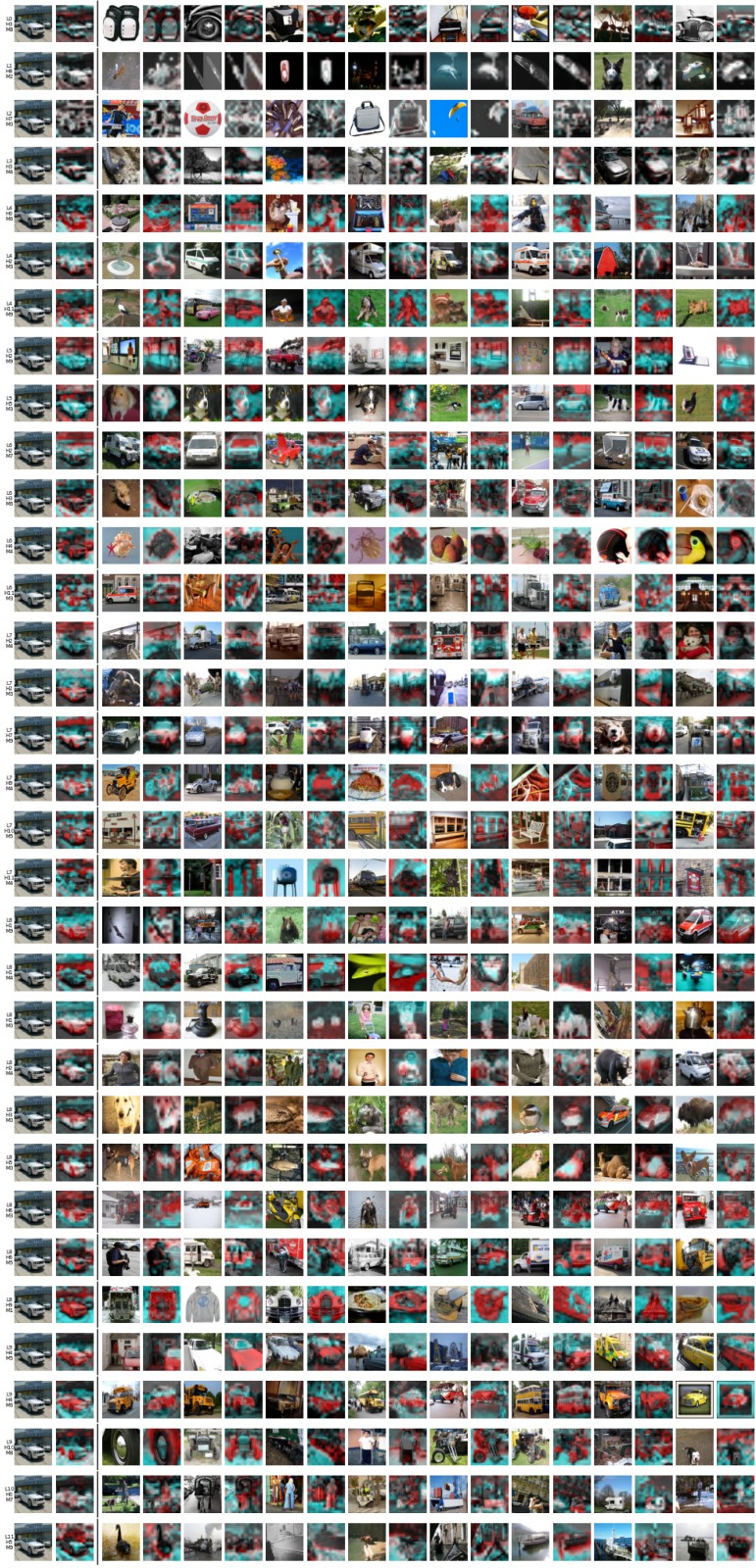

Figure S18: Singular mode maps of a car image in dino-vitb16. We hand-pick modes to show the variety of information interactions within this image. The left two columns are the original image and corresponding singular mode maps. Other columns are the top 8 images that induce the highest attention through the corresponding mode.

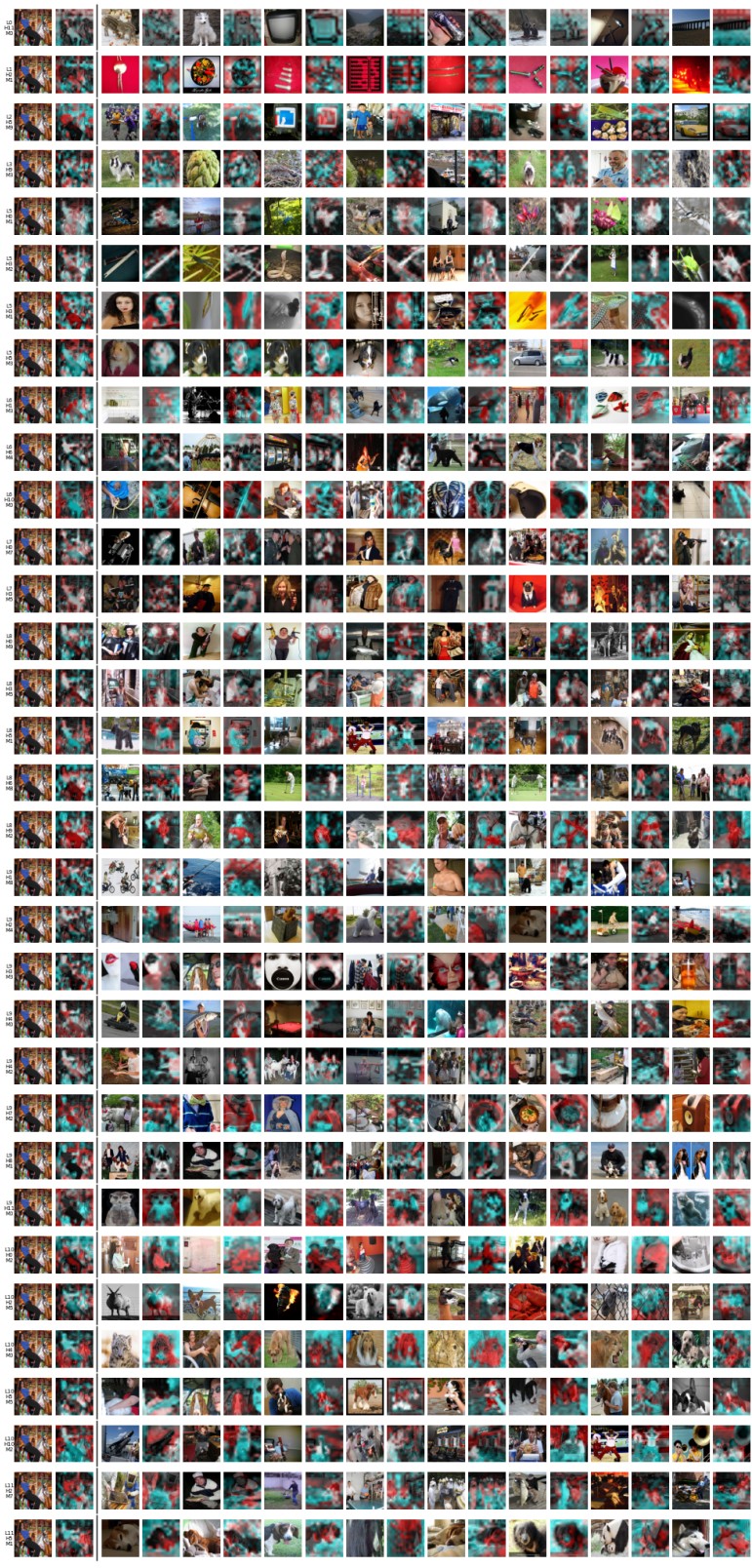

Figure S19: Singular mode maps of a human image in dino-vitb16. We hand-pick modes to show the variety of information interactions within this image. The left two columns are the original image and corresponding singular mode maps. Other columns are the top 8 images that induce the highest attention through the corresponding mode.

