# OpenReview forum: "Dissecting Query-Key Interaction in Vision Transformers"
_NeurIPS.cc/2024/Conference — NeurIPS 2024 spotlight_

### Official Review · Reviewer_Quv7 · 2024-07-10

**Soundness:** 2
**Presentation:** 2
**Contribution:** 2
**Rating:** 3
**Confidence:** 4

**Summary:**

The paper proposes SVD as a way for analyzing the interaction between the Key and Query vectors within the self-attention architecture. To this end, it measures the cosine similarity between the left and right eigenvectors of the attention score. The proposed approach evaluates the proposed mechanism on different configurations of DeiT, CLIP, DINO, and VIT. One of the findings of the paper which is the difference in the type of interaction in different layers is interesting, though to some-level could be expected given the higher abstraction level seen in later layers.

**Strengths:**

*  The paper takes an interesting approach for analyzing the interaction between queries and keys in self-attention.
*  The finding on the nature of contribution or interaction of different layers between the keys and queries  is interesting.

**Weaknesses:**

* The paper seems to be written in a haste. For example, the figures almost miss the proper x-axis labeling.
* The visualizations of the eigenvectors back-projection seems not relating properly with the claims, in most of visualization there is a diverse set of locations attended and it is not obvious how the left and right eigenvector back-projections actually relate.
* The approach seems to be specifically limited to self-attention mechanism and not obvious how it may scale to other architectures
* It is not also obvious how the scores are averaged to be visualized for different layers and over the samples in the dataset. Slight better elaboration on the dateset and the characteristic of the data would have been helping.
* While a semantically meaningful approach is proposed, the experiments are mostly focused on object-object and foreground-background interaction.
* The proposed approach could seems to be usable as a posthoc approach.

**Questions:**

* How could one connect the proposed approach to the different outputs of the model? For example, in a classification tasks, how could one propagate the interaction of Q-K vectors for different classification scores?
* Could you further elaborate on scaling this to other architectures than self-attention?

**Limitations:**

The authors discuss the limitations of the method to some degree.

---

> ### Author Rebuttal · Authors · 2024-08-06
>
> Thank you for your review and questions. We are glad you found our approach and results interesting. We believe the points you raised are either about clarifying questions or comments that do not justify the score of 3. We reply to all your questions below.
>
> > The paper seems to be written in a haste. For example, the figures almost miss the proper x-axis labeling
>
> The X-axis label of “layers” follows the convention in other papers, ordering the layers from low to high. We omitted unnecessary marks for readability and did not label the layer numbers, since different architectures have a different number of layers. We will explicitly mark that the left end is the first layer and the right end is the last layer, and add a clarifying sentence in the figure captions.
>
> > The visualizations of the eigenvectors back-projection seems not relating properly with the claims, in most of visualization there is a diverse set of locations attended and it is not obvious how the left and right eigenvector back-projections actually relate
>
> The visualizations show the projection value of a hidden embedding onto singular vector pairs. As guaranteed by SVD, ​​if a query is completely aligned with a left singular vector (red: query directions), it would only have a non-zero attention score with tokens with a non-zero projection onto the corresponding right singular vector (cyan: key directions). We found different types of interactions between the singular vectors. For example cupcake top attends to cupcake bottom; objects held by a person attends to the person holding it. Via visualizations, we find that many left and right singular vectors are related semantically, especially in higher layers.
>
> > The approach seems to be specifically limited to self-attention mechanism and not obvious how it may scale to other architectures
>
> Self-attention is one of the most important building blocks in modern deep neural networks. Though our focus in this study is vision models, our methods can be easily generalized to other modalities, cross-attention in multi-modal models, and other attention variations.
>
> > It is not also obvious how the scores are averaged to be visualized for different layers and over the samples in the dataset. Slight better elaboration on the dateset and the characteristic…
>
> For Fig 2 we average across heads and images. For Fig 3 we averaged across heads and modes, weighting modes by singular value. For Fig 5, for each mode, we find the top 5 images in the ADE20K training set that maximally activate this mode, and then we calculate the frequency with which the left singular vector and the right singular vector highlight the same object. Then this frequency is averaged per head with singular value weighting.
>
> Visualizations are the projection values of the hidden embeddings onto singular vectors. We do not average the visualizations. The visualization is per mode and image.
>
> We used ImageNet, Odd-One-Out (O3) and ADE20K datasets. O3 has been used to answer questions about saliency perception. It includes a target object, and distractors (objects similar to one another). The target has distinct properties such as shape. ADE20K is a semantic segmentation data set: all images are annotated with the objects and parts. The O3 and ADE20K sets were chosen to suit the questions we answered.
>
>
> > While a semantically meaningful approach is proposed, the experiments are mostly focused on object-object and foreground-background interaction
>
> As AI models are increasingly adopted in real-world applications, there is great interest in explainability, namely dissecting why a model behaves as it does. One direction is finding feature axes for layers of neural networks, as we discuss in the Related Work. Here we propose to examine axes interactions via query and key modes. We do not apriori look for object-object or foreground-background interactions. Rather, these, along with parts of objects, emerge from the SVD analysis. We also discuss the kinds of features that emerge across the layers. This approach could be used in future studies on different data sets and modalities, potentially resulting in other types of interactions.
>
> > The proposed approach could seems to be usable as a posthoc approach
>
> As you said, it can be considered a post-hoc approach to manipulate models and ensure safe model behavior. It is a trending field to find the internal representation of a trained transformer and then manipulate the features to avoid unsafe outputs, see Transformer Debugger, Mossing et al. 2024. Our method is very different from previous studies, 1) rather than requiring inference or training an SAE, the features are found by decomposing the weights, 2) rather than finding individual feature directions, our approach finds pairs of feature directions. We believe our approach will impact the field of model interpretability.
>
> > How could one connect the proposed approach to the different outputs of the model? For example, in a classification tasks, how could one propagate the interaction of Q-K vectors for different classification scores?
>
> Intuitively if we find several singular modes according to activation level that share semantic content, deleting these modes from the model will hinder the information processing of this semantic content; enhancing these modes will bias the model to output more relevant content to this semantic concept (see Scaling Monosemanticity: Extracting Interpretable Features from Claude 3 Sonnet).
>
> > Could you further elaborate on scaling this to other architectures than self-attention?
>
> Our approach can be applied to almost all transformer models, including self-attention and other forms of attention, such as cross-attention. For instance, in a multimodal model that includes images and audio, the query could pertain to the visual information and the key to the audio information; the modes would then highlight visual and auditory information that are attended together.

---

> > ### Comment · Area_Chair_DSCv · 2024-08-11
> > **Does the rebuttal address your concerns?**
> >
> > Dear Reviewer Quv7,
> >
> > Thank you again for your time for reviewing this paper. Could you please check if the authors' rebuttal has addressed your concerns at your earliest convenience? Thank you!
> >
> > Best regards,
> >
> > AC

---

### Official Review · Reviewer_EcHv · 2024-07-11

**Soundness:** 4
**Presentation:** 3
**Contribution:** 2
**Rating:** 7
**Confidence:** 4

**Summary:**

While previous studies on vision transformers focused on how self-attention groups relevant tokens, this paper analyzes how self-attention contextualizes tokens to understand comprehensive inter-token relationships across the entire image. To this end, this paper proposes using the Singular Value Decomposition (SVD) to analyze the query-key interaction $\textbf{W}^{\top}_q \textbf{W}_k$. Each singular vector of the query-key projection layers can be interpreted as capturing a certain type of visual semantic information. Thus, similarities between left and right singular vectors can reveal how different visual semantics interact with each other. Through extensive analysis, this paper concludes that vision transformers tend to first attend to similar tokens to form local visual semantics, and then attend to dissimilar tokens to capture global contexts of the image.

**Strengths:**

S1. **Writing**: The paper is well written and clearly motivated.

S2. **Novelty**: The proposed analysis via SVD is innovative.

S3. **Technical soundness**: The justification for using SVD is well-founded theoretically and supported by solid quantitative and qualitative empirical analysis.

S4. **Potential for broader application**: Despite being applied to image analysis, the proposed technique is generic and could potentially be applied to various domains, including video and audio understanding..

**Weaknesses:**

W1. **Familiar conclusion**: The general finding of this paper, i.e., group early and contextualize later, is already well known from many earlier papers [a,b]. This paper only reconfirms the same conclusion in an explainable manner.

W2. **Limited analysis scope**: This paper lacks analysis on different training objectives such as masked image modeling (MIM) [c,d] or masked feature prediction (JEP) [e,f]. Previous literatures [a,b] show that training objectives determine where the learner focuses; ViT with supervised training or instance discrimination self-supervision (SimCLR or MoCo) act similar as what this paper found, but ViTs trained with MIM shows that the learner still focuses on local tokens in a deeper layer. The observations in L6-8, 45-47, 138-141 might only be correct for certain training objectives.

Justification of rating: \
Despite not deriving a novel conclusion and lacking analysis on other training objectives, the proposed SVD-based analysis technique is novel, provides interpretability on query-key interactions, and is technically sound. It has potential applications in various domains.

[a] Xie et al., “Revealing the Dark Secrets of Masked Image Modeling,” CVPR, 2023.\
[b] Shekhar et al., “Objectives Matter: Understanding the Impact of Self-Supervised Objectives on Vision Transformer Representations,” ICLRW, 2023.\
[c] Xie et al., “SimMIM: A Simple Framework for Masked Image Modeling,” CVPR, 2022.\
[d] He et al., “Masked Autoencoders Are Scalable Vision Learners,” CVPR, 2022.\
[e] Assra et al., “Self-Supervised Learning from Images with a Joint-Embedding Predictive Architecture,” CVPR, 2023.\
[f] Baevski et al., “data2vec: A General Framework for Self-supervised Learning in Speech, Vision and Language,” ICML, 2022.

**Questions:**

Q1. In Fig. 3 & S3, there are slight increases in weighted cosine similarity in the final layers for all models except for the original ViTs. Does this imply that ViTs group similar tokens again in the deepest layers? Please elaborate on why this happens and what information ViTs capture in these layers.

Q2. For Fig. 4, visualizing attention maps of singular modes with fixed input images would help readers understand how ViTs adapt their focus according to the layer.

Q3. How does the diversity of visual information captured by singular vectors change across layers? In other words, how does the redundancy between singular vectors vary?

**Limitations:**

The authors adequately address both the limitation and future directions of their method.

---

> ### Author Rebuttal · Authors · 2024-08-06
>
> Thank you for your thoughtful review and suggestions.
>
> > W1. Familiar conclusion…
>
> Thank you for the references. We will include them in the “Related work” section. Our emphasis on the explainability as you say, and also on analyzing feature interactions via the query and key modes,  adds an important perspective over previous results. Our study is motivated by a seeming paradox: on the one hand visual perception requires grouping similar tokens to congregate small similar patches into cohesive bigger concepts, on the other hand visual perception also requires highlighting salience and modulating a local representation by its context. The two aspects are emphasized in different studies (see Related Work section). Our study proposes a novel analysis of Query-Key interactions via the SVD to identify if the two aspects coexist in visual transformers. Our paper clarifies they co-exist in different layers of the visual transformer, but also provides a means of explaining the types of features and interactions that emerge.
>
> We believe our approach and results complement the directions of the noted papers, by addressing feature interactions. We go beyond finding local and global aspects of attention in those papers, to understanding what types of feature interactions emerge. We find modes with semantic meaning, including attention between parts of objects, relevant objects, and foreground and background. Explainability is important for understanding existing AI models and developing new ones. As suggested by your and other reviewers' comments, our explainability approach could impact studies with different training objectives, data sets, and domains.
>
> > W2. Limited analysis scope…
>
> Thank you for pointing us to the masked model training references. We are very interested in what our approach finds for different training objectives. We have now run our simulations on a pre-trained SimMIM masked image model with a self-supervised objective, and a SimMIM model fine-tuned on imagenet classification (see Fig R2 in the attached one page pdf). A notable finding is that the SimMIM masked image model with the self-supervised objective behaves differently from the other models of Fig. 2, and has a preference for the same object in high layers. This matches the observation in the literature you suggested, that the SimMIM model has more local attention. Interestingly, the model that is fine-tuned on Imagenet classification behaves similarly to the other models in the O3 dataset experiments, highlighting the importance of the training objective. Both versions of SimMIM were taken from the official repositories. We will add these models to all of the revised paper simulations.
>
> > Q1. In Fig. 3 & S3, there are slight increases in weighted cosine similarity in the final layers for all models except for the original ViTs…
>
> This is an interesting observation. We will include a discussion on this observation in the final manuscript. With the newly added SimMIM models, we find this observation is more pronounced. The models that have this concave trend are SimMIM-pretrain, DINO, Deit, and large/huge ViT. Interestingly, most of them either have self-supervised objectives or distillation regularizations. We hypothesize that the last layer may behave differently because it is closer to the training target, and so the training objective may have more influence. This hypothesis is clearly supported by the newly added SimMIM analysis, which shows that later layer attention in the pre-trained model prefers similar features, but in the model fine tuned on image classification prefers dissimilar features. Intuitively we think the objective of reconstructing the mask requires strong local consistency, thus attention is allocated to checking the consistency of similar features. The learning objective of DINO may also require local consistency, but not as strong as reconstruction. The role of the training objective and the last layer is an interesting topic for future research. It’s worth noting that the last layer behaves intriguingly differently from other layers of the network in a number of other deep learning studies examining very different metrics and domains (e.g., Transformer Layers as Painters, Sun et al., arxiv 2024; The Unreasonable Ineffectiveness of the Deeper Layers, Gromov et al., arxiv 2024; Neural representational geometry underlies few-shot concept learning, Sorscher et al., PNAS, 2024).
>
> > Q2. For Fig. 4, visualizing attention maps of singular modes with fixed input images would help readers understand how ViTs adapt their focus according to the layer.
>
> Thanks for suggesting this interesting experiment. We now added an experiment that visualizes a single image with multiple modes. We take an image and then look for the maximally activated modes for a layer and head. Fig R1 shows results for an example dog image from the Imagenet data set and Dino transformer. We show the top 6 modes (ordered by the contribution to the attention score) for example layers and heads. The late layers capture information such as the parts of a dog or animal, and a hand with a dog. The early layers capture low-level properties.
>
> > Q3. How does the diversity of visual information captured by singular vectors change across layers? In other words, how does the redundancy between singular vectors vary?
>
> Thanks for suggesting looking at the diversity of the SVD features. The diversity of a set of features could be measured/indicated by some different metrics. As mentioned in ​​Park, Namuk, et al. "What do self-supervised vision transformers learn?”, the diversity of features could be indicated by the shape of the singular value spectrum: flat spectrum may indicate more diverse rich features. We show several singular value spectra in Fig S2. Deeper layers have flatter spectra than early layers, which may indicate deeper layers have more diverse features than early layers.

---

> > ### Comment · Reviewer_EcHv · 2024-08-11
> > **Response to Authors' Feedback**
> >
> > Thank you for addressing my concerns and questions. I appreciate the thorough analysis and the newly added results with SimMIM, which strengthen the paper. However, I have some remaining points to discuss:
> >
> > - **JEPA Results (W2)**: I respectfully request results on JEPA (Joint Embedding Predictive Architecture). JEPA, which predicts latent representations instead of pixel values, has shown good transferability in both full finetuning and linear probing setups. However, how ViTs trained with JEPA learn different representations compared to contrastive learning or MIM approaches is not well-studied. Analyzing how queries and keys interact under JEPA objective would significantly enhance the paper's contribution.
> >
> > - **Revision of Generalized Statements (W2)**: If query-key interactions vary across different training objectives, statements such as those in lines 6-8, 45-47, and 138-141 should be revised. These statements currently appear to generalize across all training objectives, but they may be limited to specific ones. Please supplement the discussion with query-key interactions observed in other objectives.
> >
> > - **Clarification of Phenomenon (Q1)**: While I appreciate the detailed explanation provided, I'm not fully convinced that it clearly and sufficiently addresses the exact reason for the observed phenomenon. This experimental trend is interesting and warrants further study. If a sufficient explanation is unavailable through this rebuttal, I strongly suggest adding related discussion and posing an open question in the main manuscript at the very least.
> >
> > Given these points, I maintain my current rating. However, I'm open to raising the score if the authors provide sufficiently clear responses to these newly added questions and concerns. Addressing these points would make the paper more comprehensive and valuable to the field.

---

> > > ### Author Response · Authors · 2024-08-12
> > >
> > > Thank you for your insightful feedback and helpful further suggestions. We will address them in the final manuscript.
> > >
> > > > JEPA Results (W2)…
> > >
> > > Thanks for suggesting adding JEPA to the analysis. We added “I-JEPA-vit-h14” into our “cosine similarity” plots. We could not attach a new figure for you at this phase, but we can tell you that the trend of “I-JEPA-vit-h14” looks similar to the “SimMIM-vit-b16-finetune”. It doesn’t increase in the last few layers like “SimMIM-pretrain”. Since the I-JEPA encoder performs well in the linear-probing setup, this indicates it is probably more similar to a classification model, which is consistent with our new analysis. The remaining question is why the self-supervised objective doesn’t cause later layers in the I-JEPA encoder to increase its attention to similar tokens. We think this objective is mainly handled by the predictor (also a transformer). To test this, we run the cosine analysis on the predictor transformer, and find the cosine similarity is around 0.3 (considered high) across layers, which is consistent with our hypothesis. Due to the time limit we don’t have the results of “I-JEPA-vit-h14” for the other analyses yet, but it won’t be a problem to finish all the analyses and add them to the final manuscript.
> > >
> > > > Revision of Generalized Statements (W2)...
> > >
> > > Thank you very much. Yes, we will change those claims and add new text on the training objective according to our discussion.
> > >
> > > We will change line 45-47 to “We identify a role of self-attention in a variety of ViTs. In many ViTs, especially those with classification training objectives, early layers perform more grouping in which tokens attend more to similar tokens; late layers perform more contextualizing in which tokens attend more to dissimilar tokens. However, this observation has some variability among models and may depend on the training objective: notably, some self-supervised ViTs tend to increase attention to dissimilar tokens in the last few layers.”
> > >
> > > We will also make the changes to line 6-8 and 138-141 in a similar way.
> > >
> > > We will add a paragraph in the discussion section: “Though we find a trend that attention changes from attending more to the similar tokens to dissimilar tokens from early layers to late layers, some ViTs have a more complex trend that increases attention to similar tokens in the last few layers (Fig. 3). Models that have this “concave” trend are SimMIM-vit-b16-pretrain, Dino models, Deit models, and huge ViT models. Most of them either have self-supervised objectives or distillation regularizations. We hypothesize that the last layers may behave differently because they are closer to the training target, and so the training objective may have more influence. We think that self-supervised objectives, such as reconstructing masked patches, require stronger consistency between tokens, and thus more attention is allocated to similar tokens in the higher layers; while the classification objective requires gathering information from different aspects of a scene, and thus more attention is allocated to dissimilar tokens. This hypothesis is supported by the cosine similarity plot (Fig. 3) of the SimMIM models, which shows in the last few layers of the pre-trained model increased attention to similar features. This matches the observation in the literature, that the SimMIM model has more local attention (Xie et al. 2022). However,  we find that the SimMIM model fine-tuned on ImageNet classification has the trend of decreased attention to similar features, similar to most of the classification models. Although I-JEPA is trained with a self-supervised objective predicting latent representations, the cosine similarity for the I-JEPA encoder does not show increased attention to similar tokens in the last few layers. The I-JEPA model is known to have excellent linear-probing performance, and thus we think it may behave more similarly to a classification model. The self-supervised objective of I-JEPA may be more apparent in the I-JEPA predictor (also a transformer). When we run the cosine similarity analysis on the predictor module instead of the encoder, we find that the cosine similarity is overall high (Supplementary Figure). The role of the training objective on internal model behavior is an interesting topic for future research.”
> > >
> > > We will also add the relevant references here for the SimMIM and I-JEPA papers.
> > >
> > > > Clarification of Phenomenon (Q1)...
> > >
> > > We agree that this trend is interesting. Our current hypothesis is that this depends on the training objective, such as the need for self-consistency in some self-supervised training objectives versus classification.  The experiments with SimMIM and I-JEPA offer interesting observations in this direction. To test these questions more completely is beyond the scope of this paper, and we will include a discussion and pose this as an open question for future research (see the text in the previous point).

---

> ### Comment · Reviewer_EcHv · 2024-08-14
>
> Thank you for the thorough analysis of the raised question. I'm satisfied with the response. I strongly suggest including new results during the rebuttal in the final manuscript. I'll raise my rating to 7.

---

> > ### Author Response · Authors · 2024-08-14
> >
> > Thank you. We will do so.

---

### Official Review · Reviewer_PNQ8 · 2024-07-13

**Soundness:** 3
**Presentation:** 3
**Contribution:** 3
**Rating:** 7
**Confidence:** 4

**Summary:**

The paper begins with the observation that self-attention in early layers of vision transformers tend to group similar objects while deeper layers focus more on gathering features from dissimilar objects or background. The paper then delves into the mathematical formulation of the attention mechanism, and reveals using SVD that the aforementioned behaviour is a result of the similarity or dissimilarity between the corresponding left and right singular vector of the projection matrices $W_q$ and $W_k$. These results are well supported by empirical evidence on numerous vision transformer backbones and rich visualisations.

**Strengths:**

1. The paper eases into the investigated problem with numerous intuitions and visualisations, which are backed up with more rigorous deductions. The narrative of the paper, as a result, is rather clear and easy to follow.

2. The paper presents an interesting finding on the behaviour of commonly used vision transformers and provides insights behind such behaviours. The finding that early layers of vision transformers group similar visual cues while deep layers extract more contextual features can help deepen the community's understanding of the dynamics within the transformer architecture.

3. The paper utilises SVD to study the product of the projection matrices for keys and queries. This methodology may be applied to understand interesting behaviours in other applications. For instance, transformer-based object detector, notably DETR, eliminates the need of non-maximum suppression by employing the self-attention mechanism amongst object queries. And self-attention was demonstrated to have suppressive behaviours.

**Weaknesses:**

1. The behaviour of deeper layers, i.e., tokens attending to dissimilar tokens, can use some more analysis. For instance, the authors stated in lines 238-239 that tokens around the fish attending to regions containing the human may add the attribute "be held" to the fish tokens. This can be easily tested by, say, manually overriding the corresponding attention scores to zero and observe if the resultant image feature has reduced cosine similarity against the language embedding "a photo of a person holding a fish". This can be easily done on a CLIP model. Analysis such as this will further deepen our understanding of transformers' behaviour, whereas simply saying deeper layers extract contextual information sounds very hollow.

2. The visualisations, such as those in Figure 4 and the appendix, only show one particular mode in one head. It would be more interesting to focus on a single image and visualise how the modes in different layers and different heads process this particular image. This builds a complete and coherent narrative around the behaviour of the model, which would include how it extracts lower-level features and the high-level features. I believe this will greatly benefit the paper.

**Questions:**

N/A

**Limitations:**

The authors pointed out the potential behavioural differences across different models and training techniques. In addition, the study in the paper was only conducted on query-key interactions, and the authors plan to investigate the role of value projection in the future.

---

> ### Author Rebuttal · Authors · 2024-08-06
>
> Thank you for your thoughtful review and suggestions.
>
> > The behaviour of deeper layers, i.e., tokens attending to dissimilar tokens, can use some more analysis…
>
> Thank you for this excellent suggestion. During the short rebuttal time frame, we conducted a preliminary test on your suggested experiment. We deleted a mode that shows the interaction between a cello and a player. We compared the output logit of an image with text input “a girl plays cello”. The logit only decreased a tiny amount (from 26.8074 to 26.8036). We think that one concept is simultaneously processed by multiple modes, so deleting one mode hardly affects the final outputs. We think this is a very interesting experiment, but due to the time limit and the depth of this question, we will not include a thorough analysis in this paper, but leave it for a future in-depth study.
>
> > The visualisations, such as those in Figure 4 and the appendix, only show one particular mode in one head…
>
> Thanks for your kind suggestion. This is a very interesting experiment which we have now conducted (see pdf Figure R1) and will be included in the final manuscript in the supplementary material. We take an image and then look for the maximally activated modes for a layer and head. Figure R1 shows results for an example dog image from the Imagenet data set and Dino transformer. We show the top 6 modes (ordered by the contribution to the attention score) for example layers and heads. We also show for each mode, the optimal Imagenet images for those modes. The late layers capture more semantic information such as the parts of a dog or animal, and a hand with a dog. The early layers capture low-level properties. We have observed that some heads have more consistency in the type of structure captured across the top modes (e.g., layer 8 head 0 shown in Fig R1 is mostly parts of object; some early layer heads are capturing frequency or position structure), while others do not in an obvious way.
>
>
> > The authors pointed out the potential behavioural differences across different models and training techniques. In addition, the study in the paper was only conducted on query-key interactions, and the authors plan to investigate the role of value projection in the future.
>
> Thanks for this comment. We agree that our current work is about finding where there is an information flow between tokens, but not about what information is passed. Studying the role of value tokens could potentially address this limitation. We are very interested in conducting such studies in future works.

---

> > ### Comment · Reviewer_PNQ8 · 2024-08-12
> >
> > Thanks for the response. I think the per-image visualisations in the attached PDF are better illustrations for the paper's argument. I would recommend adding them to the main paper.
> >
> > Regarding the modes and how they impact the logits, I suspect there may be multiple nodes exhibiting similar attention behaviours, therefore contributing to the corresponding logit concurrently.
> >
> > Nevertheless, I'm generally happy with the response, and would recommend accepting the paper.

---

> > > ### Author Response · Authors · 2024-08-12
> > >
> > > Thank you for your feedback and helpful further suggestions.
> > >
> > > > I think the per-image visualisations in the attached PDF are better illustrations for the paper's argument. I would recommend adding them to the main paper.
> > >
> > > Thank you, we will do so.
> > >
> > > > Regarding the modes and how they impact the logits, I suspect there may be multiple nodes exhibiting similar attention behaviours, therefore contributing to the corresponding logit concurrently.
> > >
> > > Thank you, this is also our interpretation, and an interesting direction for future work. We will add this to the limitations section as a future research direction.

---

### Official Review · Reviewer_LpZi · 2024-07-14

**Soundness:** 3
**Presentation:** 4
**Contribution:** 4
**Rating:** 8
**Confidence:** 5

**Summary:**

This paper proposes a new analysis framework to dissect the potential bottom mechanism of query-key interactions in Vision Transformers (ViTs)  from the perspective of singular value decomposition. Several phenomena are presented via extensive quantitative and qualitative results, which leads to the basic conclusion that earlier layers in ViTs tend to conduct grouping among similar tokens and deeper layers in ViTs are more likely to take on the role of contextualization to connect dissimilar tokens.

**Strengths:**

1.This paper investigates the working mechanism of query-key interactions in the popular ViTs, which is of great importance to enhance the explainability of transformer models but has been rarely explored before.

2.A novel conclusion is given that the self-attention layers are not limited to conduct grouping among similar tokens. In deeper network layers, they also perform something like contextualization over dissimilar tokens to extract higher-level semantics.

3.In addition, a new analytical tool for ViTs is provided in this work to visualize the attention preference of different self-attention layers, i.e., calculating the inner product between visual tokens and singular vectors of query/key weight matrices. This could benefit future research to conduct visualization analysis to facilitate the model development.

**Weaknesses:**

1.Analysis conducted in this work is mainly restricted to the ImageNet dataset, where the images are more focused on relatively simpler scenes and objects. To make the conclusions more general, experiments on samples with more complex visual scenes are more helpful.

2.The ViT models investigated in this work are mainly pretrained vision models with general training objectives. It could be more comprehensive and more intetesting to see how the ViT models fine-tuned under specific downstream tasks will behave.

**Questions:**

Although the interaction mechanism between queries and keys in ViTs are explored in this work, how the value projection layers take effect and how the query-key itneraction matrices coordinate with the value tokens to influence the output features remain unclear. I suggest the authors to conduct further studies in future on these points.

**Limitations:**

The current limitations have been discussed by the authors in the manuscript.

---

> ### Author Rebuttal · Authors · 2024-08-06
>
> Thank you for your thoughtful review and suggestions.
>
> > 1.Analysis conducted in this work is mainly restricted to the ImageNet dataset…
>
> We are interested in applying this approach to more complex visual scenes and other different datasets and domains in the future.
>
> > 2.The ViT models investigated in this work are mainly pretrained vision models with general training objectives…
>
> Thank you. We are very interested in these questions. Following the suggestion of the reviewers, we have now analyzed a pre-trained mask reconstruction model SimMIM that has a self-supervised objective, and its variant that is fine-tuned on Imagenet classification. We find interesting differences across the tasks (see Fig R2 in the pdf). For example, in the O3 dataset experiments, the SimMIM masked model attends more to similar objects in late layers (relating to work that suggests attention in this model is more localized), whereas the fine-tuned classification model behaves more similarly to ViT attending more to different objects or the background for high layers. This observation is consistent with previous studies that show SimMIM focuses on local features (Park, Namuk, et al. "What do self-supervised vision transformers learn?").
>
> > Although the interaction mechanism between queries and keys in ViTs are explored in this work, how the value projection layers take effect and how the query-key itneraction matrices coordinate with the value tokens to influence the output features remain unclear. I suggest the authors to conduct further studies in future on these points.
>
> Thanks for your kind suggestion. We agree that our current work is about finding where there is an information flow between tokens, but not about what information is passed. Studying the role of value tokens could potentially address this limitation. We are very interested in conducting such studies in future works.

---

> > ### Comment · Reviewer_LpZi · 2024-08-10
> >
> > Thanks for the authors' rebuttal and additional experimental results. After reading the authors' response, my previous concerns have been adequately addressed, so I decide to raise my rating to 8.

---

> > > ### Author Response · Authors · 2024-08-12
> > >
> > > Thank you for your feedback.

---

### Author Rebuttal · Authors · 2024-08-06

Dear Reviewers and AC,

Thank you for your helpful reviews. We appreciate that reviewers thought our SVD approach for finding query-key interactions is novel/interesting, and that our methodology has applicability to other domains and applications.

We appreciate the thoughtful suggestions. In response, we have run new analyses, which we believe has improved our paper, namely:

**(i) Visualizing a single image with multiple modes:** Two reviewers suggested we run our analysis on a single image to see how the modes change across the heads and layers. This is a good idea and we have now run this analysis (see examples in Fig R1 of the attached pdf). We will include the new simulations in the revised paper as part of the supplementary material.

**(ii) Influence of the training objectives via newly added SimMIM transformer analysis:** Two reviewers noted the interest in downstream tasks and other training paradigms, which we agree, and we appreciate the pointer by one of the reviewers to the masked image model literature. We have therefore run our simulations on the SimMIM pretrained model, and on a version of the SimMIM which was fine-tuned for Imagenet classification. Interestingly, the fine-tuned model behaved similar to the ViT model, but the self supervised masked training model behaved quite differently, relating to previous findings that the SimMIM masked image model is more localized in its attention. We will include the new simulations in the revised paper (see Fig R2 in the one page pdf).

We also reply individually to all other comments and suggestions by the reviewers.

---

### Comment · Area_Chair_DSCv · 2024-08-09

Dear Reviewers,

Thank you very much again for your valuable service to the NeurIPS community.

As the authors have provided detailed responses, it would be great if you could check them and see if your concerns have been addressed. Your prompt feedback would provide an opportunity for the authors to offer additional clarifications if needed.

Best regards,

AC

---

### Decision · Program_Chairs · 2024-09-25

**Decision:**

Accept (spotlight)

**Comment:**

Most of the reviewers acknowledged the importance of the task investigated in this paper, novelty of the proposed approach and conclusion, the broader application of the proposed approach to other Transformer models other than ViTs, and clear writing of the paper. Major concerns of the reviewers have been address in the authors' rebuttal. The reviewer Quv7 still has skepticism and would like to see more analysis after the rebuttal. However, the AC found this criticism lacks concrete evidence and elaboration.

Overall, the AC found this paper of high quality and recommend to accept it as a spotlight.